# Alpine-style nappes thrust over ancient North China continental margin demonstrate large Archean horizontal plate motions

Yating Zhong [1], Timothy Kusky [1,2,4 ✉], Lu Wang [1,4 ✉], Ali Polat [1,3], Xuanyu Liu[1,5], Yaying Peng[1,5], Zhikang Luan[1,5], Chuanhai Wang[1,5], Junpeng Wang [1] & Hao Deng[1]

Whether modern-style plate tectonics operated on early Earth is debated due to a paucity of definitive records of large-scale plate convergence, subduction, and collision in the Archean geological record. Archean Alpine-style sub-horizontal fold/thrust nappes in the Precambrian basement of China contain a Mariana-type subduction-initiation sequence of mid-ocean ridge basalt blocks in a 1600-kilometer-long mélange belt, overthrusting picritic-boninitic and island-arc tholeiite bearing nappes, in turn emplaced over a passive margin capping an ancient Archean continental fragment. Picrite-boninite and tholeiite units are 2698 ± 30 million years old marking the age of subduction initiation, with nappes emplaced over the passive margin at 2520 million years ago. Here, we show the life cycle of the subduction zone and ocean spanned circa 178 million years; conservative plate velocities of 2 centimeters per year yield a lateral transport distance of subducted oceanic crust of 3560 kilometers, providing direct positive evidence for horizontal plate tectonics in the Archean.

[1] State Key Lab for Geological Processes and Mineral Resources, and Center for Global Tectonics, School of Earth Sciences, China University of Geosciences, Wuhan 430074, China. [2] Three Gorges Research Center for Geohazards, China University of Geosciences, Wuhan, China. [3] School of the Environment, University of Windsor, Windsor, ON N9B 3P4, Canada. [4] These authors jointly supervised this work: Timothy Kusky, Lu Wang. [5] These authors contributed equally: Xuanyu Liu, Yaying Peng, Zhikang Luan, Chuanhai Wang. ✉email: tkusky@gmail.com; wanglu@cug.edu.cn

One of the greatest challenges facing Earth Sciences today is obtaining a better understanding of early Earth's tectonic regime, and how it developed from Hadean accretion through the Archean and Proterozoic to form the life-sustaining tectonically active surface we live on today[1,2]. Perhaps most pressing is determining for how long the present-style mobile lid comprised of numerous rigid plates moving relative to each other on a globally-linked network of divergent, convergent, and transform boundaries has been in operation, or if other hypothesized modes of planetary heat loss were dominant in Archean times, such as plume/sagduction modes, represented by vertical movements[3,4]. Documentation of geological relationships that unambiguously demonstrate large horizontal motions of plates on Earth in the Archean is critical to test if plate tectonics operated in early times.

Here, we report unequivocal hallmark evidence for Archean subduction and large-scale horizontal plate motions, culminating in the emplacement of a series of subhorizontal fold nappes with regional-scale overturned limbs, onto a formerly distant continental margin. These structures are considered diagnostic indicators of horizontal plate translations in modern orogens. We document this smoking gun evidence for early horizontal plate motions from the Archean basement of the North China Craton (NCC), and note the remarkable similarity to the Cenozoic Alpine nappes formed by plate convergence and collision[5,6]. We show that rocks in the nappes formed in the forearc of a large (>1600-km long) intra-oceanic island-arc system, and were emplaced, after circa 178 Ma of horizontal transport, over a continentally derived sedimentary sequence of metapelites, quartzites, and marbles deposited unconformably over an older gneissic terrane which includes rocks with ages ranging from 3.8 to 2.5 Ga, with a major crustal growth event at ~2.7 Ga and a crustal reworking event at 2.5 Ga. The lower stack of nappes includes circa 2698 ± 30 Ma island-arc tholeiites and high-MgO, high-SiO$_2$, low-Ti picritic-boninitic volcanic rocks, hallmark signatures of Mariana-type subduction initiation in extant and Phanerozoic arcs[7,8]. The latter is intruded by circa 2692 ± 35 Ma evolved trondhjemitic dikes, similar to mid-upper crustal tonalities-trondhjemites of intra-oceanic arcs[9,10], with an age difference from subduction initiation to arc magmatism similar to that in Mariana-type subduction-initiation events[7,8,11], This forearc subduction-initiation sequence is overthrust by an orogenic mélange belt, that extends for more than 1600 km along strike, containing a wide variety of blocks including MORB-type ophiolitic rocks[12,13], in a highly-deformed metasedimentary matrix. The entire package was then intruded by 2500 Ma undeformed granitic dikes, marking the end of deformation and nappe emplacement[13]. The mélange separates the forearc fold/thrust nappe sequence from an intra-oceanic arc complex[13,14] that also extends for 1600 km along strike, forming the 2700–2500 Ma Wutai-Fuping arc of the Central Orogenic Belt of the NCC[13,14]. Together with recently documented paired metamorphic belt relationships[15], ultra-high pressure (UHP) mineral inclusions[16], and forearc-ophiolitic remnants[12], our new documentation of large-scale horizontal transport of Alpine-style fold nappes in the Central Orogenic Belt represents some of the best evidence yet for the operation of Phanerozoic-style plate tectonics on the Archean Earth.

The remarkable high degree of similarity between orogenic structure, zonation, and contained rock units and their relationships in the Archean Central Orogenic Belt and the Alpine and Appalachian systems, clearly formed by plate tectonics, suggests that plate tectonics is the most reasonable mechanism by which to interpret this Archean sequence. Therefore, we use this clear example of subhorizontal Alpine-style forearc nappes thrust over a continental margin in the Archean to make quantifications of the translational length-scales of Archean plate motions. We use the time interval from subduction initiation and formation of picrite-boninite series rocks at 2698 Ma, to the best-estimate of the time of initial emplacement of the nappes over the passive-margin-bearing continent at 2520 Ma, for a measure of the life cycle of the ocean between the arc and the continent of 178 Ma (Supplementary Note 1). If we then use a conservative convergence rate[17] of 2 cm/yr (20 km/Ma) between the arc/forearc sequence and the passive margin over which it was emplaced, we obtain a minimum length of subducted oceanic crust of 3560 km; i.e., the horizontal transportation distance of the nappe sequence from where it formed in the Archean ocean, to where it was emplaced over the passive margin. This represents the first geologically well-constrained estimate of minimum distance of horizontal plate tectonic translations in the Archean.

## Results

**Geological and tectonic classification**. The Central Orogenic Belt (COB) of the Neoarchean North China Craton (NCC) (Fig. 1a) is a 1600-km-long Neoarchean collisional orogen formed between an intra-oceanic arc terrane in the west, and the ancient continental Eastern Block in the east[12,13,18–20]. The zone between the arc and craton includes dissociated remnants of passive-margin sequences, complex mélange belts with blocks of Neoarchean MORB-tholeiites, and ophiolitic fragments[12,13,18,21,22], UHP mineral inclusion[16], and preserved paired metamorphic relationships[12,15], representing one of the best-documented Archean classic subduction/accretion/collisional orogens on Earth.

The Zanhuang complex is located on the central easternmost margin of the COB[13] (Fig. 1a), and is subdivided into three domains by structures and lithologies[21,23–25] (Figs. 1b and 2). In this contribution, we show that these domains relate to their vastly different origins: (1) The Eastern Zanhuang Domain (EZD) consists of mainly intermediate to felsic gneisses and migmatites of an older continental fragment of the Eastern Block of the NCC, and is overlain by a 2.7–2.5 Ga passive continental margin platform sequence (marble-siliciclastic sediments). The gneisses are part of a regional terrane that includes rocks with ages ranging from 3.8 to 2.5 Ga[13], with younger events reflecting partial melting during ~2.5 Ga tectonism[13,25] (Fig. 2); (2) The Western Zanhuang Domain (WZD) is comprised of 2.70–2.55 Ga island-arc affinity associations including tonalitic gneisses, granites, and amphibolites[24], correlated with the Wutai/Fuping arc in the COB[20,24–26]; and (3) The Central Zanhuang Domain (CZD), containing an intensely deformed Neoarchean tectonic mélange, and a fold nappe/thrust belt comprised of metapelites, metapsammites, metabasalts including pillow lavas, metagabbros, and rare ultramafic rocks, intruded by a circa 2.5 Ga granitic pluton and undeformed pegmatites[23,25]. In this integrated work based on detailed field structural-lithological mapping, we show that horizontal plate motions exceeding 3500 km can be demonstrated in the Archean.

**Structural-lithological analysis of the Zanhuang fold nappes**. The Zanhuang fold nappe/thrust belt is a highly-sheared and intensely deformed tectonostratigraphic zone in the Central Orogenic belt. At 1:100,000 scale the zone is only mappable as an incoherent chaotic highly mixed mélange belt[25] (Fig. 1b), but our new detailed (1:666) structural and lithological mapping has revealed a new coherent sub-unit of fold nappes (profile A-B, Fig. 1b) within the mélange belt that exhibits critical details of the evolution of the belt that directly relate to the style of Archean plate tectonics. The nappes include units of metasedimentary (shale, greywacke, and chert) and meta-volcanic origin, deformed

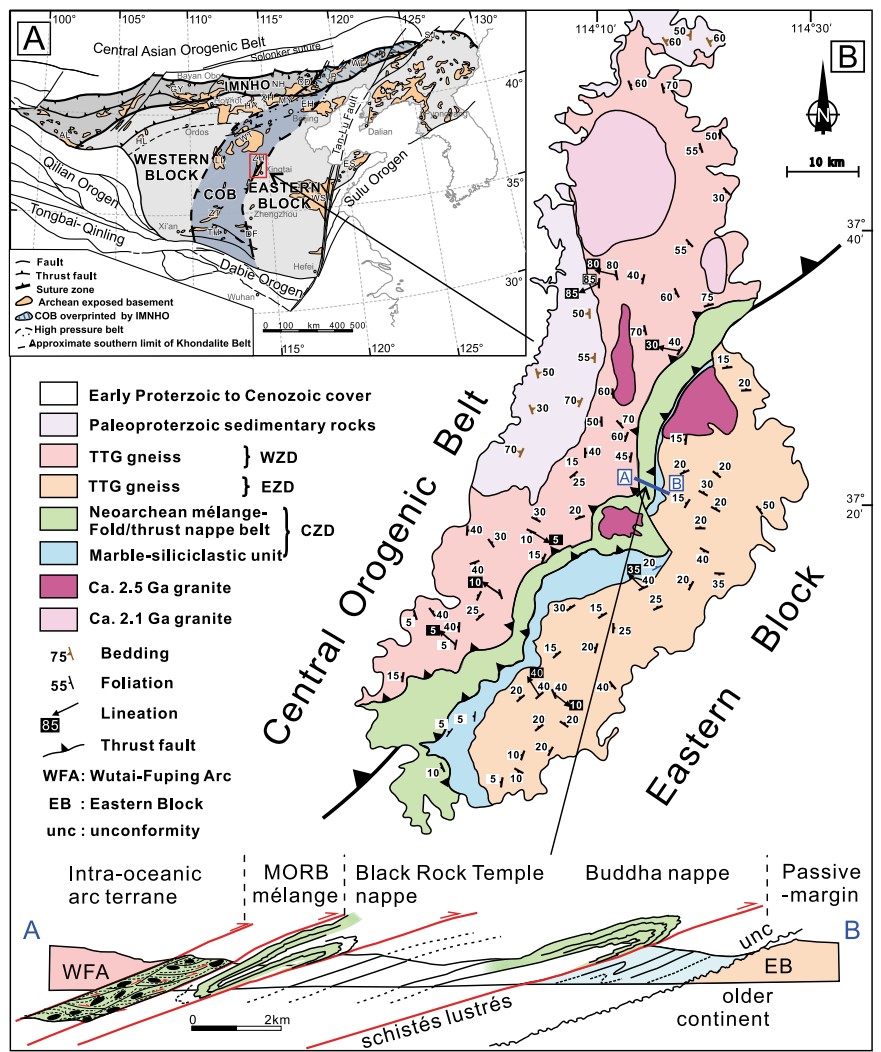

**Fig. 1 Geological map and tectonic setting of the area studied. a** Map showing main Precambrian tectonic divisions of the North China Craton (modified from Kusky et al.[13]). Red box shows location of Zanhuang Complex, illustrated in (**b**); (**b**) Map showing the main tectonic zones of the Zanhuang Complex, with simplified profile (no vertical exaggeration) along fold nappe/thrust belt (**a**, **b**), showing the relative positions of the Eastern Block (EB) basement and unconformably overlying passive margin. The forearc upper-plate units include the Buddha nappe, thrust over the passive margin along a major thrust (suture) marked by the *schistes lustrés,* and this is succeeded structurally upwards by the Black Rock Temple nappe, which is overthrust by the Wutai/Fuping arc (WFA) system along the MORB-bearing ophiolitic mélange zone. WZD Western Zanhuang Domain, EZD Eastern Zanhuang Domain, CZD Central Zanhuang Domain. Map (**b**) modified from Wang et al. and Kusky et al.[12,13,16,21,23].

into strongly flattened fold structures (Fig. 2). The main-phases of deformation (D$_1$ and D$_2$) pre-date circa 2.50 Ga post-orogenic intrusions[23,25], and are related to an imbricated thrust and nappe complex (Figs. 2, 3). The folds extend ~15–20 km across strike, and 100 km along the length of the Central Zanhuang Domain[21,23,25]. Our mapping and quantitative structural analysis (Fig. 2) reveal that individual nappes are typically hundreds of m to ~1-km thick, extending along strike for several to tens of km, until their fold closures meet those of overlying nappes. Foliations formed during D$_2$ nappe-style folding dip shallowly (15–40°) NW, toward the hinterland/ancient sea (Fig. 2, stereographic projections). The nappes are tight-to-isoclinal fold structures, overturned with SE-vergence towards the foreland, with their axial surfaces subparallel to the overturned limbs (Figs. 2, 4a, b, d, e). The overturned limbs are the sites of thrust faults, where individual units are repeated, forming thrust duplex structures (Fig. 3a). Some show well-developed sheath folds in 3-dimensions, represented by strong curvilinear hinge lines (Fig. 2, stereographic projections), and elliptical cross-sections, indicating

extremely high shear strains (Fig. 3c). Sense-of-shear criteria (asymmetric folds, rotated porphyroclasts/blasts, mica fish, duplex structures), are ubiquitous (Figs. 3, 5d, e), indicating that the fold nappe/thrust belt underwent intense, non-coaxial ductile deformation with upper nappe units gliding to the southeast along thrust planes lubricated by greasy mica-rich metapelites and related rocks of the *schistes lustrés*, described below. These main structural elements are locally overprinted by younger tectonic and related metamorphic features[27].

Several large (<1×~10–20 km) nappes contain metamorphosed picrite-boninite-series volcanics and island-arc tholeiite basalts (IAT) (Fig. 2). The lowermost allochthonous nappe is named "Buddha nappe" (after a nearby village name). It is a large-scale, recumbent fold, the overturned limb of which is ~100–150-m thick and highly attenuated due to stretching. The amphibolite-grade Buddha nappe is resting upon a clean-cut thrust fault (Figs. 1, 2) decorated by highly schistose and sheared dominantly mica-quartz-rich metapelites, ranging from 3-cm to 3-m thick. We colloquially refer to the sheared metapelites and related

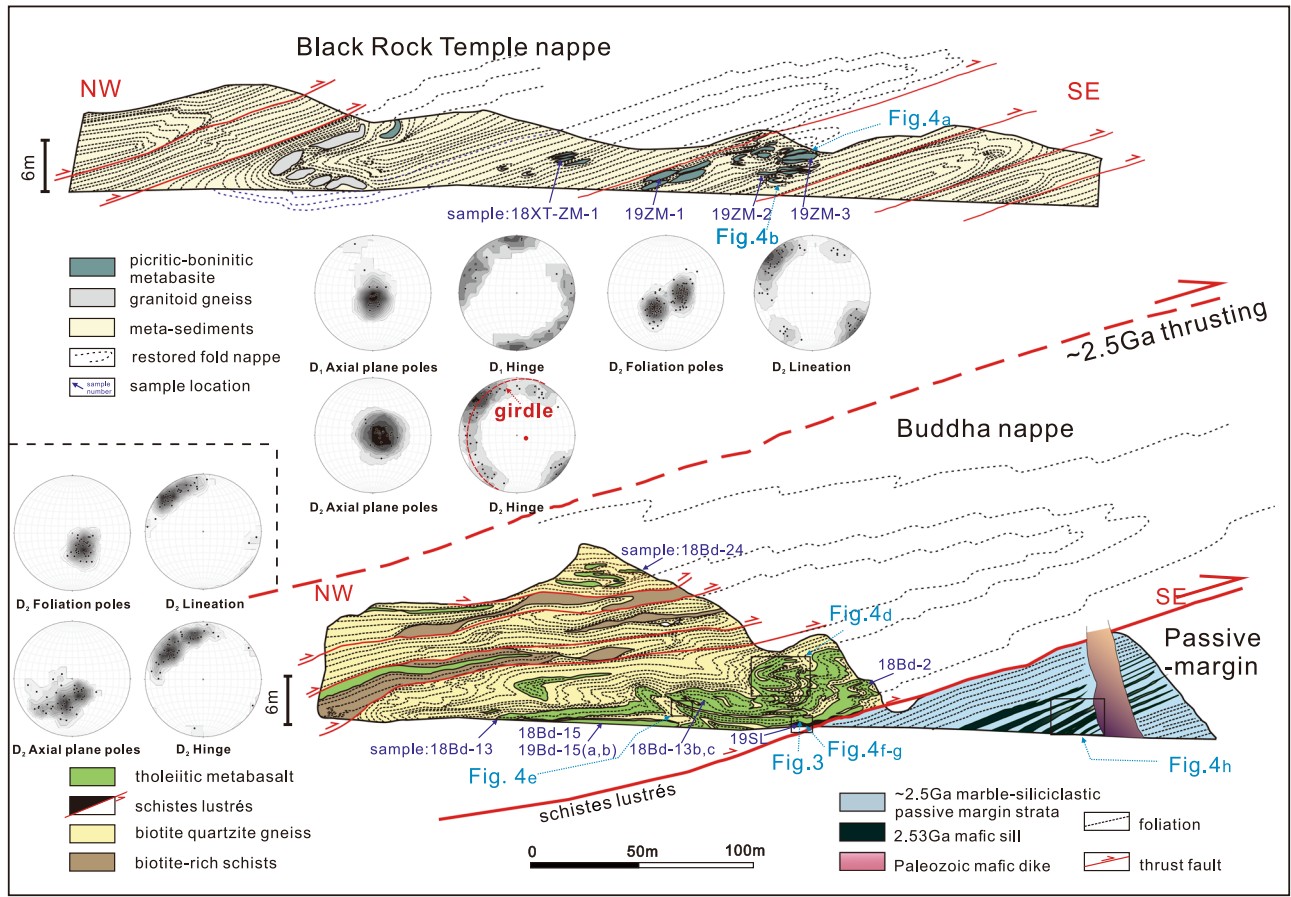

**Fig. 2 Profile of Black Rock Temple and Buddha nappes (both named after local names); lower-hemisphere equal area stereographic projections of $D_1$ and $D_2$ axial surfaces, fold hinge lines, and lineations.** Structural data for Black Rock Temple nappe is above dashed line, for the Buddha nappe, below line. Mapping was performed at a scale of 1:666.

metasediments along this thrust surface as *schistes lustrés*, because of their close similarity to examples from the Pennine Alps and Corsica[5]. Amphibolite-grade nappes of the entire domain are detached from the deeper substrate and thrust over the lower-grade greenschist facies *schistes lustrés* and autochthonous sequence of marble-siliciclastic passive continental margin strata. The layer of *schistes lustrés* (Figs. 2, 3d–f, 4f, g) is a high Al, Mg, Fe, low Ca, mica-rich metamorphosed clastic shiny tectonite or phyllonite, which has relatively low shear resistance that facilitated transportation of the allochthonous material. The top of Buddha nappe is marked by another highly schistose thrust, which in turn is overthrust by additional large nappes including the Black Rock Temple nappe with similar dimensions to Buddha nappe, but containing picritic-boninitic lavas described below (Figs. 1, 2).

**Mapping, sampling, petrography, and geochronology.** Key samples were selected and collected based on their structurally well-constrained positions, and carefully filtered for effects of weathering and alteration (Supplementary Note 2); sample locations are shown in Fig. 2. From the mapping and structural analysis, major nappe units (the upper Black Rock Temple nappe, and lower Buddha nappe), that contain distinct suites of metabasic rocks with associated intrusive and sedimentary assemblages were delineated. Descriptions of the lithotectonic assemblage from each discrete nappe are reported separately

below, since they are separated by major ductile shear zones (faults).

The Black Rock Temple nappe is mainly comprised of units of picrite-boninite metabasites within a metasedimentary (grey-wacke and shale protoliths) dominated sequence. The folded picrite-boninite metabasites are crosscut by an undeformed felsic (trondhjemitic) dike (Fig. 4c), constraining the age of the primitive picrite-boninite series volcanic rocks that it cuts.

Primitive picrite-boninite series volcanic rocks of the Black Rock Temple nappe are dark-green hornblendite with massive to foliated textures, crosscut by felsic dikes (Fig. 4a–c). The picrite-boninite series rocks are metamorphosed into amphibolites dominantly composed of large grains of hornblende with minor plagioclase and epidote, preserving relict igneous textures (Fig. 5a). Minor accessory minerals include zircon, magnetite, and titanite.

Felsic dikes (Fig. 4c) crop out as white- to gray-colored trondhjemite with plagioclase and/or quartz phenocrysts in the Black Rock Temple nappe. They consist of plagioclase (Pl, 55–60%) and quartz (Qtz, 5–30%), with minor biotite (Bi, ~4%) and potassium feldspar (Kfs, <4%), and accessory minerals including apatite and zircon. They preserve relict original subhedral plagioclase phenocrysts demonstrating their magmatic origin (Fig. 5b). Biotites form dusty brown matrix minerals and collections of tiny flakes along foliation planes. The trondhjemite dike that crosscuts the picrite-boninite unit yields a U-Pb (zircon) age of 2692 ± 35 Ma (Supplementary Note 1).

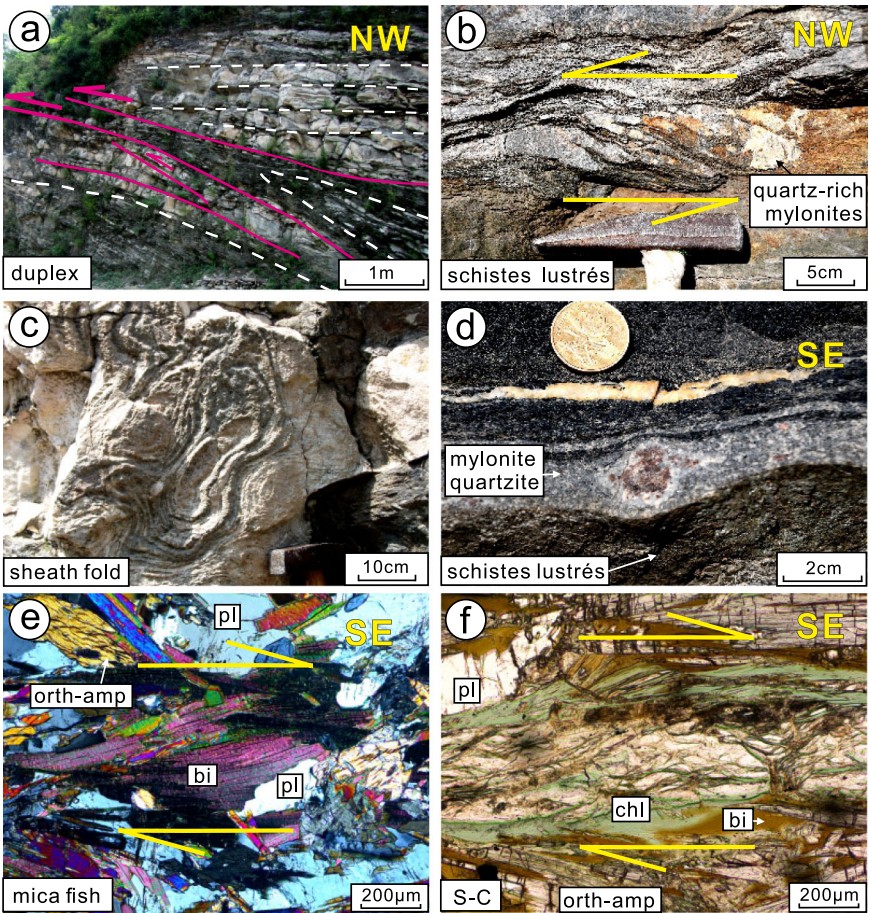

**Fig. 3 Representative deformation features of the Zanhuang fold nappe/thrust belt. a** Roof and floor thrusts in a duplex structure, with incipient horses bounded by linked thrusts, and an incipient nappe climbing over the floor ramp; **b** small-scale nappes in recrystallized quartz-rich mylonites along the upper contact of the *schistes lustrés*, mimicking the larger-scale geometry of the belt; **c** sheath fold in marble, looking along hinge to the NW; **d** contact of the *schistes lustrés* (bottom of photo) marked by a recrystallized quartz-rich mylonite with asymmetric σ-shaped garnet porphyroblast (top to the SE sense of shear), above which is the tholeiite of the Buddha nappe; **e** photomicrograph of the *schistes lustrés*. showing asymmetric mica fish, top to the SE sense of shear; **f** photomicrograph of *schistes lustrés*, with S-C fabric defined by micas, showing top to SE sense of shear. bi biotite, chl chlorite, orth-amph orthoamphibole, pl plagioclase.

The Buddha nappe includes layers of tholeiitic metabasalt folded within less competent quartz-biotite gneisses and biotite-rich schists. The lower limb of the Buddha nappe is a shear zone, comprised of a relatively low shear resistance layer of high-Mg, Al, and Fe, low-Ca, mica-rich dominantly metasedimentary shiny schist, which we describe as *schistes lustrés*[5,28,29]. Tholeiitic metabasalts of the Buddha nappe are dark-green, garnet porphyroblast-bearing foliated amphibolites (Fig. 4d, e), with a medium-grained prismatic blastic texture. The tholeiites are chiefly composed of garnet (Grt, 5–8%), biotite (Bi, 0–12%), amphibole (Amp, 15–36%), plagioclase (Pl, 30%), and quartz (Qtz, 8–27%), with minor K-feldspar (Kfs), chlorite (Chl), and rutile (Ru) in a medium- to fine-grained matrix. Amphibole and biotite exhibit a strong preferred orientation (Fig. 5c). Garnet forms rotated porphyroblasts with sigmoidal and spiral-shaped inclusion trails making spectacular snowball garnet structures (Fig. 5d, e), providing evidence for syntectonic crystallization during D2 non-coaxial shearing. Two tholeiitic basalt samples yielded U/Pb (zircon) crystallization ages of 2698 ± 30 Ma, and 2699 ± 24 Ma, and metamorphic ages of 2453 ± 45 Ma and 1852 ± 29 Ma (Supplementary Note 1), respectively, the former of which represents syntectonic zircon recrystallization during D$_2$, while the latter represents a younger overprinting tectonothermal event. Peak metamorphic mineral assemblages include amphibole, garnet, plagioclase, and quartz and give amphibolite facies metamorphism conditions.

Rocks we refer to as *schistes lustrés* demarcate the lower limb of the Buddha nappe, and shear zones separating other major fold nappes of the belt. They comprise greyish-green colored, metamorphosed clastic shiny schists (Fig. 4f, g) characterized by abundant flaky-prismatic crystalloblastic chlorite (Chl, 25%), biotite (Bi, 15%), muscovite (Mus, 5%), and orthoamphibole (Orth-amp, 20%), with plagioclase (Pl, 20%) and quartz (Qtz, 20%), in a medium- to fine-grained matrix (Figs. 3e, f, 5f). Accessory minerals include apatite and zircon. The *schistes lustrés* were metamorphosed into shiny schists during deformation as shown by syntectonic garnet, quartz, and mica growth (Fig. 3e, f), similar to phyllitic *schistes lustrés* in the Pennine Group (Simplon-Ticino nappes) and blueschist facies *schistes lustrés* in Alpine Corsica[5,28,29]. The unit consists of dominantly metasedimentary rocks, and detrital zircons from the *schistes lustrés* yield a maximum depositional age of 2536 ± 43 Ma showing that no rocks younger than this were contributing detritus, and a metamorphic age of 2455 ± 26 Ma, showing that the deformation was over by then (Supplementary Note 1). This age is consistent with other data we have obtained from the mélanges[12,13,21] and

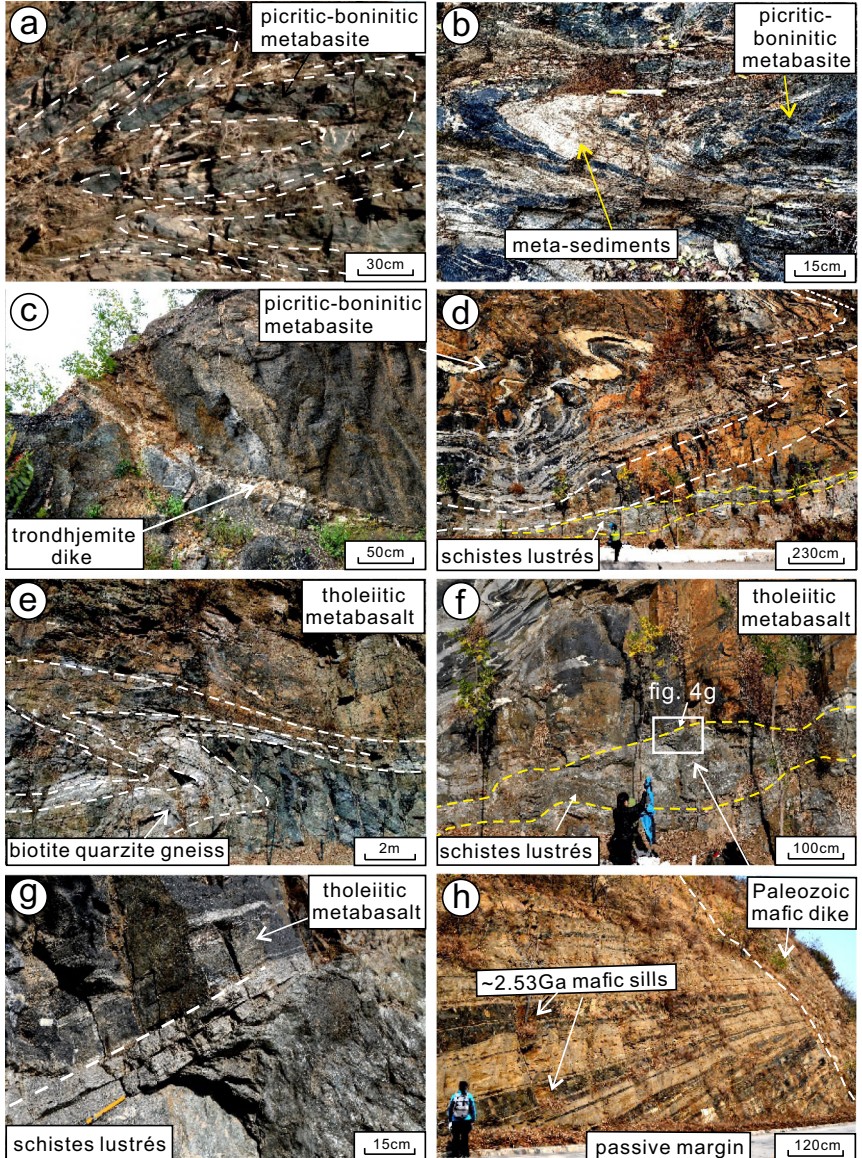

**Fig. 4 Field photographs illustrating field relationships and lithological characteristics in the Buddha and Black Rock Temple nappes, Zanhuang Complex. a** Folded amphibolites of picritic-boninitic composition from the picrite-boninite unit of the Back Rock Temple nappe; **b** tight to isoclinally folded picritic-boninitic metabasite (dark layer) with magnetite-quartzite (light layer, interpreted as meta-hydrothermal chert); **c** detail of picritic-boninitic metabasites crosscut by trondhjemite dike; **d** recumbent nappe with parasitic folds of tholeiitic basalt composition resting upon the thrust fault defined by the *schistes lustrés* layer. Note that this is an oblique view with highly curved hinge surfaces that vary from the core to the outside of the fold structure: **e** tholeiitic metabasalts and biotite-quartzite gneiss disposed in a large-scale recumbent fold in the Buddha nappe (one layer is outlined with white dashed line); **f** the variably-thin *schistes lustrés* layer, marking the thrust (suture) between the allochthonous IAT tholeiites of the Buddha nappe (above) and the autochthonous siliciclastic metasediments of the passive-margin sequence below; **g** detail of the contact (along yellow pencil) of the *schistes lustrés* thrust zone with overlying allochthonous Buddha nappe, represented by folded tholeiitic basalts. Pencil (yellow) is on the top of the *schistes lustrés* thrust zone. **h** Footwall metasediments of the passive-margin sequence in the Buddha nappe intruded by 2.53 Ga mafic sills and Paleozoic dolerite dike.

allochthonous units of the Central Orogenic belt, showing that they were emplaced regionally between 2520 and 2500 Ma, then cut by undeformed granitic dikes and pegmatites at 2500 Ma, with growth of metamorphic zircon rims continuing through 2480–2460 Ma[12,15,18,21,23,30]. Based on our thin section observation, the peak metamorphic mineral association of Pl, Qtz, Chl, Bi, Mus, and Orth-amp provide a rough estimation of metamorphic condition of greenschist facies.

The age data is important to constrain the timing of nappe emplacement. The minimum depositional age (MDA) of 2520 Ma means that sedimentation and thrusting are younger than that age, and the cross-cutting 2500 Ma granites means thrusting is older than that age. From these data, we suggest that the Zanhuang nappes were emplaced between 2520 and 2500 Ma, and we use the conservative MDA 2520 Ma age as the time of initial emplacement for our calculations.

**Whole-rock major and trace element geochemistry**. Metabasites from the Buddha and Black Rock Temple nappes are strongly metamorphosed and deformed, yet their textures and petrography demonstrate magmatic protoliths. Zircons (Supplementary Note 1) show oscillatory zoning and monotonous age values,

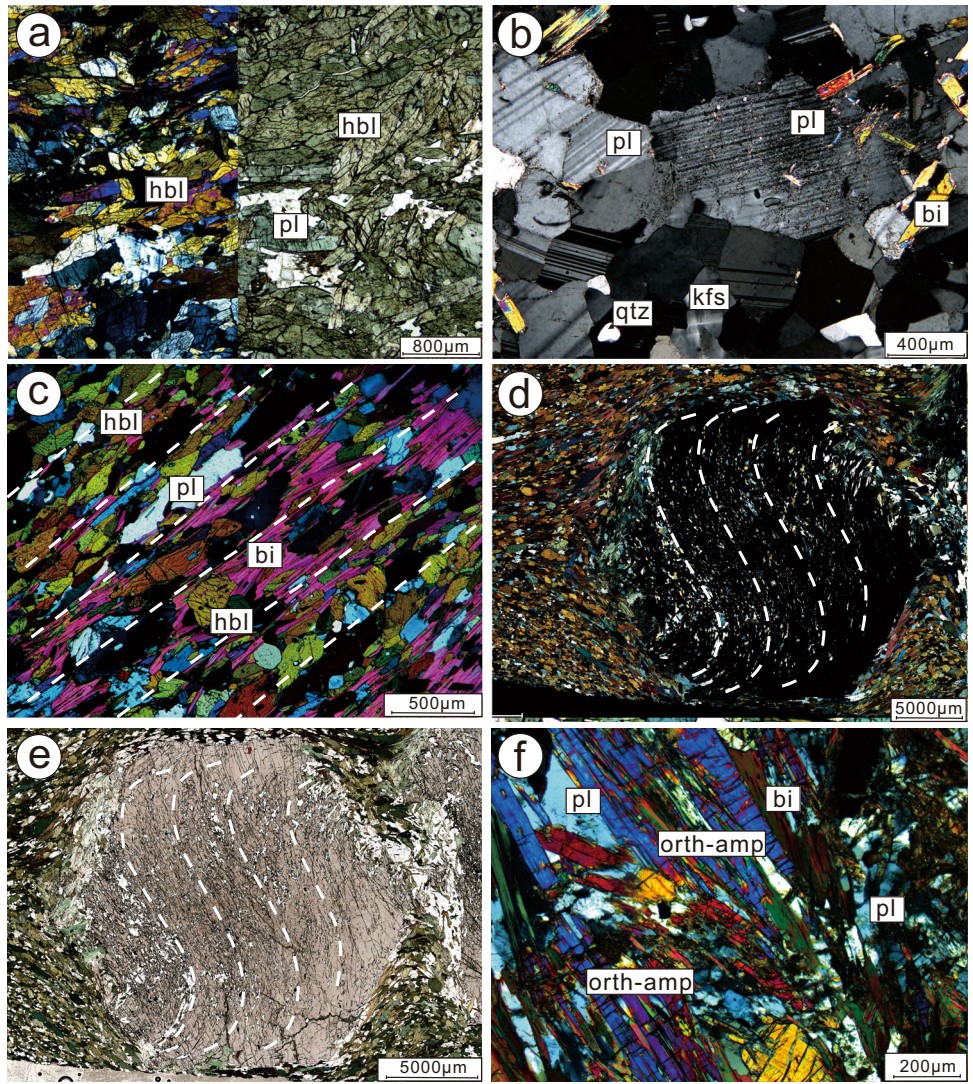

**Fig. 5 Photomicrographs showing representative mineralogy and microstructures of typical lithologies. a** The picritic-boninitic metabasite is predominately composed of large grains of hornblende with minor plagioclase; microphotograph shows a continuous view but left half is under cross polars and right half is under plane light; **b** microphotograph of the trondhjemite dike. The relict of original subhedral plagioclase phenocrysts corroborates the magmatic genesis for the precursor; **c** tholeiitic basalt is strongly foliated and chiefly composed of garnet (shown in **d**, **e**), biotite, amphibole, plagioclase, and quartz; **d**, **e** garnet forms rotated porphyroblast with sigmoidal and spiral-shaped inclusion trails making beautiful snowball garnet structures, which are traditionally interpreted as evidence for syntectonic crystallization; **f** *schistes lustrés* consisting of flaky chlorite, biotite, and prismatic orthoamphibole with plagioclase and quartz. bi biotite, chl chlorite, grt garnet, hbl hornblende, ilm ilmenite, orth-amph orthoamphibole, pl plagioclase.

supporting a magmatic protolith. After assessing for element mobility (see Supplementary Note 2), we differentiate two main geochemical types of metabasites that both plot in the ortho-amphibolite domain on the MgO-CaO-FeO$_{tot}$ diagram[31] (Fig. 6a), and follow a tholeiitic differentiation trend (Fig. 6b) corroborating their magmatic protoliths.

Zanhuang picrite-boninite series rocks are characterized by high MgO (12.90–15.19 wt%), CaO (8.96–12.44 wt%), and SiO$_2$ (48.08–54.42 wt%), depleted aluminum with Al$_2$O$_3$ of 5.59–9.02 wt% and low TiO$_2$ (0.41–0.48 wt%) contents (Supplementary data 2). Their high Mg# (60.9–75.4) with high compatible trace elements (Cr = 963–1450 ppm) point to their compositional similarity with primary melt characterized by Mg#~70, Cr >1000 ppm[32]. The boninite samples are plotted on the revised IUGS classification diagram[33] for high-Mg rocks (Fig. 6c), showing that the primitive lavas fall near the boundary between the picrite and boninite fields. Figure 6d shows a SiO$_2$ vs.

MgO variation diagram, with our primitive samples falling in the picrite-boninite series defined by Bénard et al.[34].

Figure 7a, b shows the picritic-boninitic series rocks plotted on variation diagrams with analyses from Phanerozoic and Archean boninites and picrites[9], showing that our samples fall in the range of typical picrite-boninite series rocks of all ages, using multi-element discriminants. On chondrite- and primitive mantle-normalized trace element diagrams (Fig. 7c, d), picritic-boninitic series rocks display enriched LREE (La/Sm$_{cn}$ = 1.25–3.63) and continuous depletion from MREE to HREE (Gd/Yb$_{cn}$ = 1.07–2.08) with moderate to weakly negative Eu (Eu/Eu* = 0.51–0.86) anomalies; strong LILE enrichment and HFSE depletion with negative Ti, Nb, and Hf anomalies are conspicuous. Some samples show loss of Th, due to alteration in shear zones (Supplementary Note 2).

The tholeiitic metabasalts series contain low to moderate SiO$_2$ (44.03–51.39 wt%), variable Al$_2$O$_3$ (13.35–17.08 wt%), moderate

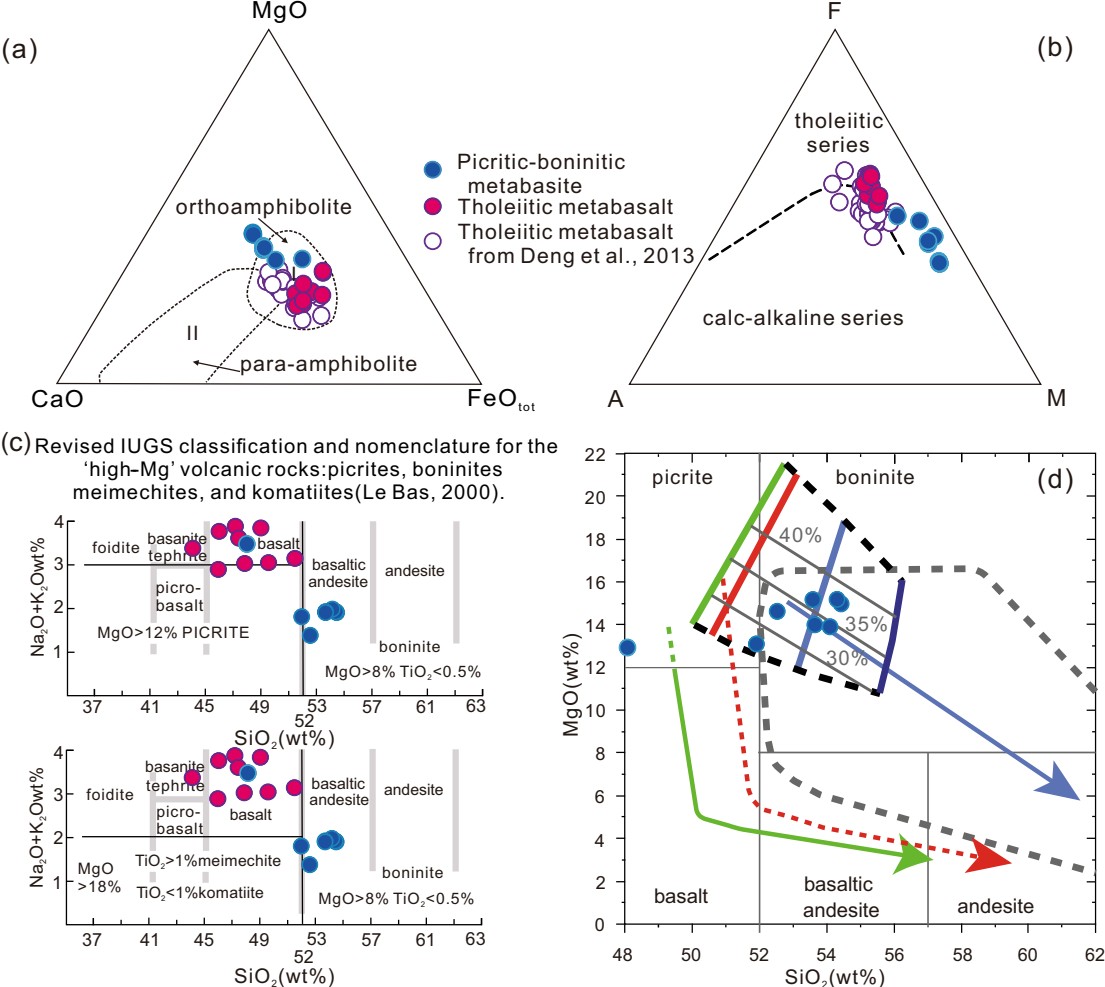

**Fig. 6 Discrimination diagrams constraining protolith of the metabasite. a** MgO-CaO-FeO$_{tot}$[31] is for the amphibolites, with domain I referring to ortho-amphibolite and domain II to para-amphibolite. Our data is in blue for the picrite-boninite series, red for the tholeiites, plotted with data from other Zanhuang basalts (open circles) from Deng et al.[26]; **b** AFM diagram[35] showing that the metabasalts plot along the tholeiitic trend; **c** SiO$_2$ vs. alkalis classification scheme and nomenclature[33] for high-Mg volcanic rocks including picrite, basalt, and tholeiite, with the Zanhuang samples plotting in the fields of basalt (for the tholeiites) and near the boundary of the picrite-boninite fields; **d** SiO$_2$ vs. MgO plot[34] with our primitive samples plotting in the picrite-boninite series. Dark gray contour outlines field of Izu-Bonin bonitites. Thick colored lines represent the compositional range of melts modeled by Bénard et al.[34] from samples of the hybrid mantle wedges of the Kamchatka and Bismark forearcs, whereas the green, red, and blue lines with arrows represent modeled liquid lines of descent[34] from picritic and boninitic parental melts, and the gray lines indicate the melting percent.

CaO (6.18–10.29 wt%), and MgO (6.97–9.66 wt%) with Mg# (46.4–54.9), TiO$_2$ (1.01–2.05 wt%) (Supplementary data 2). The samples show tholeiitic affinity on the AFM diagram[35] (Fig. 6b). The metabasalts have chondrite-normalized REE patterns characterized by slightly enriched LREEs relative to HREEs, with (La/Yb)$_N$ values of 2.43–3.29, and weak to no Eu anomalies with Eu/Eu* = 0.82–1.11 (Fig. 7e, f). The primitive mantle-normalized incompatible trace elements patterns display enrichment of large ion lithophile elements (LILE) and light rare earth elements (LREE), and negative anomalies in some high field strength elements (HFSE) including Nb, Ta, and Ti. When alteration effects are taken into account (Supplementary Note 2), some samples are seen to have lost Th, and no Eu anomaly remains, consistent with their classification as tholeiites.

**Petrogenesis of the Zanhuang subduction-initiation sequence.** The studied picritic-boninitic samples yield a primitive boninitic chemistry with high MgO (>12 wt%) contents at high SiO$_2$ (>52 wt%, with one exception of 48 wt%), with low TiO$_2$ (<0.5 wt%) contents and an important depletion of Ti relative to REE, as

well as elevated Mg# (60.9–75.4), Ni = 152.5–467 ppm, Cr = 963–1450 ppm, on the basis of the IUGS definition and volcanic geochemistry classification[33,36,37].

In contrast, typical island-arc picrites as defined by the IUGS[33] have SiO$_2$ < 52%, MgO > 12%, total alkali content of 3%, and have positively fractionated HREE patterns with (Gd/Yb)$_{cn}$ > 1.5, and Al$_2$O$_3$/TiO$_2$ ratios that are chondritic to subchondritic[9]. As shown by the melting experiments of Bénard et al.[34], and data in Polat and Kerrich[9], picrites and boninites form part of a magmatic series extending to high-Mg andesites and possibly sanukitoids that form under hot subduction conditions. Their composition reflects the P-T melting conditions, mantle wedge condition prior to melting (i.e., whether fertile, or refractory from prior melting events), variable contributions of volatiles from the slab, and the aggregation of melt fractions from the source peridotites. Multi-element discriminant plots and variation diagrams (Figs. 6, 7; Supplementary data 2) show that the Zanhuang picritic-boninitic primitive metabasites have mixed picritic-boninitic characteristics and fall within the established picrite-boninite series.

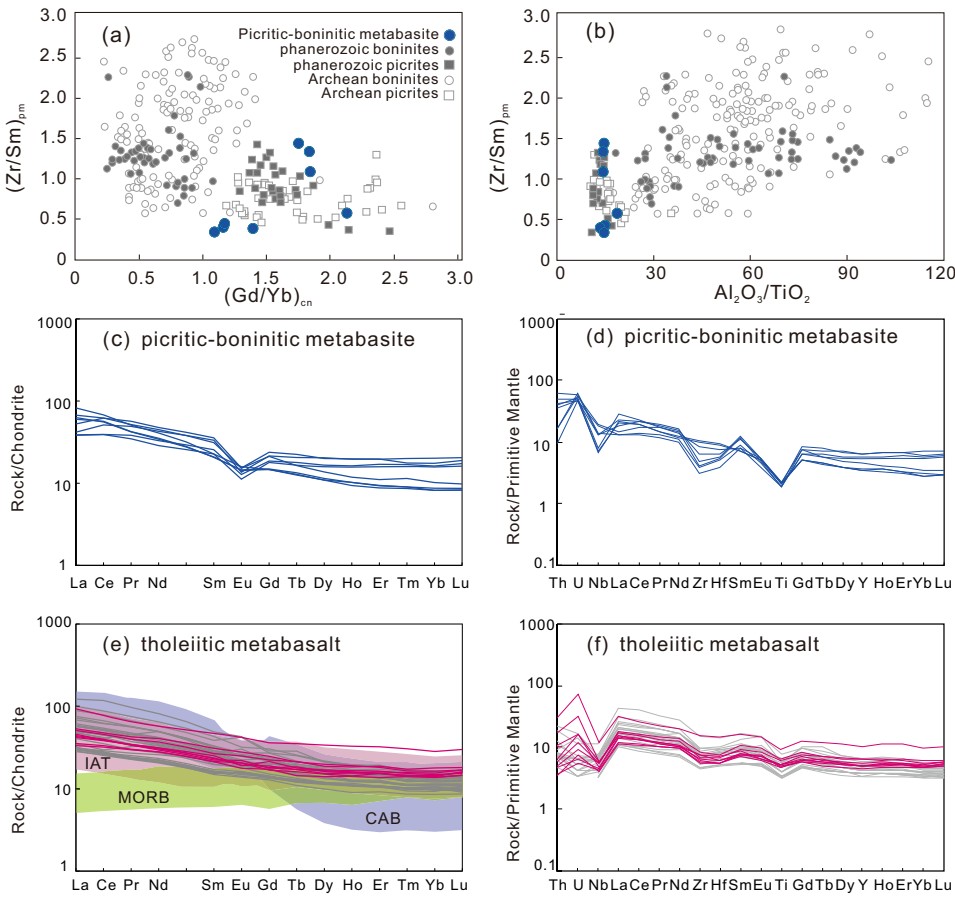

**Fig. 7 Multi-element discriminant plots and variation diagrams of picritic-boninitic-tholeiitic series rocks. a**, **b** Variation diagrams for a global dataset of Archean and Phanerozoic picrites and boninites[9] with our samples (blue) falling within the range for most elements and ratios. **a** Shows $(Gd/Yb)_{cn}$ vs. $(Zr/Sm)_{pm}$, **b** shows $Al_2O_3/TiO_2$ vs. $(Zr/Sm)_{pm}$. **c**–**f** Chondrite- and primitive mantle-normalized diagrams for picritic-boninic and tholeiitic basalts. Normalization values from Sun and McDonough[60]. The gray lines represent reported IAT samples from the Zanhuang complex[26]; the fields for MORB-IAT-CAB (Middle Oceanic Ridge Basalt-Island arc tholeiite basalt-Calc-alkaline arc basalt) data are collected from the (1) 2640–2522 Ma MORB-IAT-CABs in the Fuxin terrane, western Liaoling[61,62]; (2) 2657–2523 Ma MORB-IAT-CABs in the Shaqiaohe-Zhuzhangzi-Shangying terrane, Eastern Hebei[63–65]; (3) 2658–2550 Ma IAT in Sanhuangzai Terrane, Dengfeng complex[66]. Our data is shown by the solid blue and red lines (data is in Supplementary Note 2).

The REE patterns and Primitive Mantle-normalized incompatible trace element patterns are characterized by LREE enrichment over HREE and the LILEs are markedly enriched relative to neighboring elements (Fig. 7e, f). The elevated abundance of LREE and LILE is inconsistent with its initially depleted refractory mantle source resulting from previous melt removal, implying a subsequent re-fertilization in the mantle source before melting to form the picritic-boninitic series magmas. Eliminating significant crustal contamination (Supplementary Note 2), these distinctive geochemical features are interpreted to result from metasomatism of the mantle source region by hydrous fluids or melts in a suprasubduction setting. In the Th/Yb–Nb/Yb proxy[38] (Fig. 8a) for recycled crustal components and selective Th and Nb additions by slab-derived fluids, samples plot above the mid-ocean ridge basalt (MORB)–OIB array (Fig. 8a) supporting such subduction-related tectonic process.

Possible melting conditions that generated compositions similar to the Zanhuang picritic-boninitic series rocks were obtained by Bénard et al.[34], who sampled mantle xenoliths from the sub-arc mantle wedges of the active Kamchatka and Bismark arcs. Through experimental studies along with melting parameterizations, they produced picritic-boninitic melts with compositions similar to ours (Fig. 8c, d) under conditions of 20–45% melting of the refractory hybrid mantle wedge at 1340–1360 °C, with 1.4–3.3% water. These conditions are best met during hot subduction during subduction initiation, when the mantle wedge is undersaturated, since it has not yet been fully hydrated by long-term subduction, after which it will produce typical arc-type calc-alkaline basalt-andesite-dacite-rhyolite (BADR) magmas[9,34].

The island-arc tholeiitic (IAT) and transitional mid-ocean ridge tholeiitic basalts (MORB) are distinguished from the picrite-boninite series rocks by higher $TiO_2$, $Al_2O_3$, and lower MgO, $Fe_2O_3$, CaO abundance, and evident Nb-Ta, Zr-Hf anomalies (Supplementary data 2; Fig. 7e, f). They exhibit slight enrichment in LREE, with $(La/Yb)_N$ values of 2.4–3.2 and $(La/Sm)_N$ values of 1.6–1.9, which are chemically similar to enriched mid-oceanic ridge basalts (E-MORB) (Fig. 8a). Ruling out significant contamination (Supplementary Note 2), their distribution along an enrichment trend on a Th/Yb–Nb/Yb diagram (Fig. 8a) is displayed by their displacement to higher Th/Yb values from the MORB-OIB diagonal array, suggesting that the lavas interacted with a subduction-derived component (altering the original Th values; Supplementary Note 2). This is in agreement with the subduction-related geochemical proxies of the samples, which display remarkable enrichment of LILE but depletion of HFSE, due to generation from mantle metasomatized by slab-derived melt addition or fluids released from the dehydration of a subducting oceanic slab[38–40].

The tholeiitic basalts plot mostly in the MORB field or the transitional field between MORB and IAB on the $DF_1$–$DF_2$

 

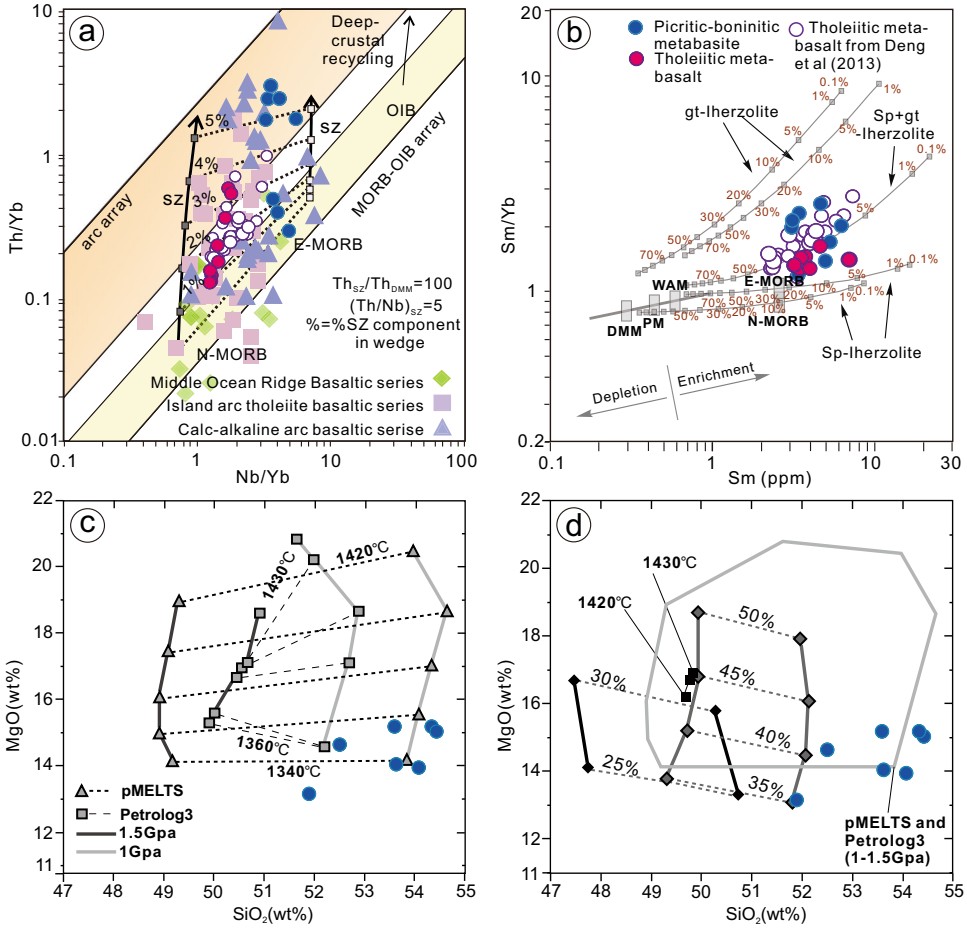

**Fig. 8 Petrogenesis discriminant diagrams of the Zanhuang picritic-boninitic-tholeiitic series rocks. a** Th/Yb versus Nb/Yb, tectonic discriminant diagrams[38] for the metabasaltic samples. Picritic-boninitic series are represented by blue circles whereas tholeiitic basalts are represented by red circles; the MORB-IAT-CAB data are collected from Fuxin-Chaoyang terrane, western Liaoning[61,62]; Shaheqiao-Qinglong-Shangying terrane, eastern Hebei[63–65]; Zanhuang complex[26]; Sanhuangzai domain in Dengfeng complex[66]. **b** Plots of Sm/Yb vs Sm diagram[67], on which melt curves correspond to degrees of partial melting for a given mantle source by the non-modal batch melting equations[68]. PM (Primitive Mantle), N-MORB (Normal Middle Ocean Ridge Basalt), E-MORB (Enriched Middle Ocean Ridge Basalt), and WAM (Western Anatolian Mantle). **c, d** Results of experimental studies along with melting parameterizations by Bénard et al.[34], where they produced picritic to boninitic melts with compositions similar to ours (Supplementary data 2). Our data is plotted with blue symbols, along the extensions of the temperature and melt percentages from Bernard et al.'s experiments[34], and from this, we infer that our samples formed under conditions of 20–45% melting of the refractory hybrid mantle wedge at 1340–1360 °C, with 1.4–3.3% water. One anomalous sample was discounted due to the deviation from the main plotting field.

diagram[41] (Fig. 9), which denotes their similarity with island-arc basalts, and also reflects transitional features between MORB and arc affinity geochemical characteristics, implying their protoliths were products derived from large degree partial melting of metasomatized (previously depleted) mantle wedge sources, in a juvenile island-arc setting (Fig. 9d).

## Discussion

Picrites and boninites are members of a rare group of "hot sub-duction" volcanic rocks, including boninites, picrites, low-Ti tholeiites, adakites, high-magnesian andesites, and Nb-enriched basalts[9], found almost exclusively in (1) forearcs of extant intra-oceanic arcs and (2) ophiolite complexes that represent former forearc settings, spatially and temporarily linked to embryonic arc volcanism following intra-oceanic subduction initiation. Boni-nites and picrites are therefore taken as a proxy for recognizing and interpreting episodes of subduction initiation in the geologic record[7,8,42]. Examples include those in the (1) well-known Cen-ozoic Izu-Bonin-Mariana (IBM) forearc and the Tonga Ridge, associated with intense mantle depletion at shallow depths in the

proto-arc wedge as subduction initiation proceeded[43] (i.e., less than c. 2 GPa or c. 60 km); and (2) forearc ophiolites (e.g., Cape Vogel boninite in Papua New Guinea; Cenozoic Nepoui boninite in the Caledonia ophiolitic nappe) thrust over continental frag-ments during arc-continental collision[44,45].

Here, we document well-preserved picrite-boninite series pri-mitive magmas that typify IBM- (or simply Mariana) type sub-duction initiation[7,8] and IAT-bearing nappes, overthrust by a locally ophiolitic and MORB-bearing mélange zone, then an intra-oceanic arc terrane, all emplaced over a passive margin in the Neoarchean Central Orogenic Belt. The Zanhuang boninitic fold nappe/thrust belt is thus perhaps the best Archean example of a well-documented forearc subduction-initiation sequence, where the structures, rock types, and temporal and spatial rela-tionships can be directly compared with their Phanerozoic counterparts for a better understanding of Archean subduction dynamics[1,2].

Chemical, petrographic, structural, temporal, and spatial rela-tionships between the picrite-boninite suite and IAT basalts suggest that they formed in a proto-forearc setting in an intra-oceanic arc system. A well-characterized type-forearc magmatic

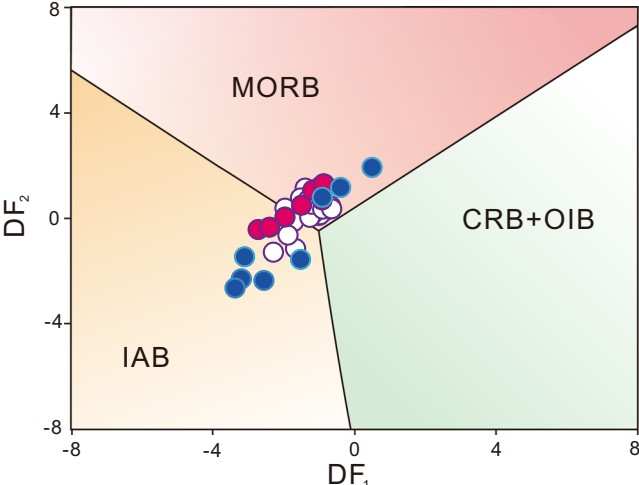

**Fig. 9 Tectonic discrimination diagrams for the metabasites.** Log-transformed immobile trace element tectonic discrimination adopted from Agrawal et al.[41] (symbols as for Fig. 8); $DF_1 = 0.3518 \log_e (La/Th) + 0.6013 \log_e (Sm/Th) - 1.3450 \log_e (Yb/Th) + 2.1056 \log_e (Nb/Th) - 5.4763$; $DF_2 = -0.3050 \log_e (La/Th) - 1.1801 \log_e (Sm/Th) + 1.6189 \log_e (Yb/Th) + 1.2260 \log_e (Nb/Th) - 0.9944$. MORB Mid-Ocean Ridge Basalts, OIB Ocean Island Basalts, IAB Island Arc Basalts, CRB Continental Rift Basalts. Our samples plot along the distinctive SSZ mantle enrichment trend reflecting slab-derived fluid metasomatism, which is also explicitly documented in the great amount of ultramafic-mafic volcanic rocks derived from both Cenozoic intra-oceanic arcs and Archean equivalents, the former (MORB trend) is represented by Tonga, Mariana, Kermadec, Kuril, Aleutian, and Scotia arcs; the latter (transitional MORB to IAT) preserved in the Eoarchean (3.8–3.7 Ga) Isua greenstone belt, Mesoarchean (3075 Ma) Ivisaartoq–Ujarassuit greenstone belt and (2970 Ma) Fiskenæsset complex[69] and Neoarchean Zanhuang tectonic mélange[26].

succession has been established on the basis of some of the best-preserved and studied nascent forearc ophiolites globally[7,8,42,46], culminating in the 'subduction initiation rule'. Such magmatic successions (Fig. 10a) are expressed by a chemo-stratigraphic variation that form MORB-like lavas, derived from decompression melting of upwelling fertile asthenosphere caused by forearc extension during trench retreat (i.e., rollback of subducted slab leads to upper-plate extension in the proto-forearc setting, and upwelling of underlying mantle), to typical SSZ (suprasubduction zone)-like lavas (with magma types including picrite-boninite-IAT basalt), resulting from continued melting of a progressively depleting, harzburgitic residue with increasing inputs of slab-derived fluids/melts from the evolving subduction zone as the sinking lithosphere descends[7,8,47,48]. Recently, it has been proposed that this type of subduction-initiation sequence is not represented in the geological record until the Cambrian at about 520 Ma[49], and therefore that modern-style plate tectonics did not emerge unit then. The documentation of the Zanhuang circa 2698 Ma Mariana-type subduction-initiation sequence, and other Archean examples[13,18] puts the time of documentable Mariana-type subduction initiation, and the operation of modern-style plate tectonics some 2.2 billion years older than the assembly of the Gondwana Supercontinent, and may have been even much older[10,50].

In the Zanhuang fold nappe/thrust belt and adjacent MORB mélange, the SSZ-affinity picrite-boninite-IAT-MORB basalt association is exposed as an on-land remnant of allochthonous ancient oceanic crust incorporated into the Archean Central Orogenic Belt and thrust over a passive continental margin sequence (Fig. 10b, c). This association provides clear evidence for

the SSZ-type ophiolitic magma origin. The chemistry specifies that the IAT basalts with HFSE depletion formed from partial melting of mantle sources that experienced previous melt extraction; the picritic-boninitic basalts that exhibit pronounced Ti depletion and high V and Sc implies high degrees of partial melting of refractory mantle sources that underwent multi-stage melt extraction[36]. Both the picritic-boninitic rocks and IATs show variable enrichments in LILE and LREE, suggesting that the previously depleted mantle sources underwent metasomatic enrichment from subduction-derived fluids/melts in the supra-subduction mantle wedge[37,51]. In addition, the scattering of the picrite-boninite-IATs in the fields transitional from MORB-like affinity to IAT/arc affinity on tectonic discriminant plots (Fig. 9) consistently plot in the array from MORB → IAT trends in our geochemical synthesis using data from 2.7 to 2.6 Ga MORB and IAT in the COB; Fig. 8a), reflecting the compositional evolution of the forearc magmatic system by progressive source depletion (e.g., lower HFSE concentration) coupled with increasing super-imposed slab-fluid-derived metasomatism (e.g., higher abundance of LREE, LILE). Such an evolution would result from normal tholeiitic and calc-alkaline arc basaltic magmatism as subduction progresses, signifying the establishment of a mature subduction zone[7,51], which is also justified by geochemical synthesis shown in Fig. 8a.

Subhorizontal nappes overlying ductile high-strain zones in orogenic belts have long been recognized as indicative of plate tectonic-driven horizontal transport of the nappes over under-lying units, either an autochthonous basement, or additional nappe units, exemplified by the Alpine system[6]. With the advent of the plate tectonics theory, researchers began to recognize that nappes, including the Taconic allochthons[52] and Bronson Hill anticlinorium nappes[53] of the Appalachians, and the Pennine nappes of the Alps[5,6], were transported many hundreds of kilometers from their sources. Later, when plate tectonics demonstrated that arcs and exotic terranes could travel across oceans, it became apparent that many allochthonous nappes were non-native to the continent they were emplaced upon, and had traveled thousands of kilometers from where they formed, across ocean basins as forearc ophiolites, arcs, and rifted continental fragments in orogens ranging in age from Cenozoic to Proterozoic[54,55].

The Archean Zanhuang picritic-boninitic and IAT nappes, formed in a forearc setting during subduction initiation at 2698 Ma, and were emplaced on a distant continental margin at 2520 Ma, 178 Ma later. The subduction-initiation age is inferred by the crystallization age of forearc-affinity metabasalt; whereas the emplacement age of nappes is constrained by the maximum depositional age (2520 Ma) of detrital zircon in *schistes lustrés*, which represents the oldest age of the nappes emplacement and provide a minimum value (178 Ma) to the period from birth to death, accordingly.

Experimental constraints[34] show that rocks with similar compositions form at 1340–1360 °C, with 20–45% shallow melting of a slab-fluid fluxed harzburgitic mantle wedge, with water (volatile) concentrations of 1–3% (undersaturated), conditions best met during hot subduction initiation before the mantle wedge is saturated by slab-derived fluids. The elevated temperatures could reflect either the higher Archean mantle potential temperature[3], or that subduction initiation was along a ridge.

Since the picritic-boninitic magmas formed in an intra-oceanic arc system, it is reasonable to assume that the arc and its associated subduction system were active for 178 Ma, consistent with the duration of arc magmatism in the Wutai/Fuping arc[13,14], similar to the duration of the modern Japanese island arc[56]. Applying a conservative estimate for plate velocity of 2 cm/yr during subduction that formed the Wutai/Fuping arc after the

**(a) ~2.7 Ga subduction initiation**

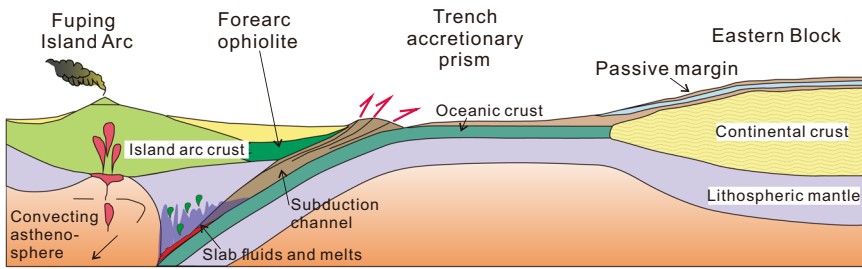

**(b) 2.68-2.55Ga Subduction-accretion and arc magmatism**

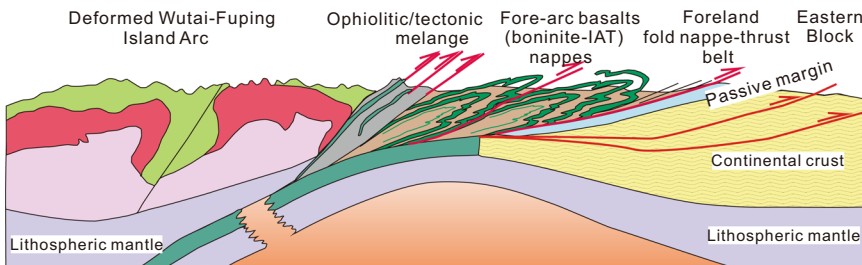

**(b) 2.50-2.45 Ga Arc-continent collision and metamorphism**

**Fig. 10 Schematic cartoon of tectonic evolution.** Tectonic evolution of the Zanhuang fold nappe—thrust belt from **a** subduction initiation at circa 2700 Ma, **b** arc magmatism during 2692–2550 Ma, to **c** emplacement of the boninite and Buddha nappes over the passive margin on the Eastern Block at 2520 Ma, and metamorphism and foreland basin sedimentation continuing until 2450 Ma. **a** Adapted from Whattam and Stern[7] with permission from [Springer Customer Service Centre GmbH] [Springer] [Contributions to Mineralogy & Petrology] [The 'subduction initiation rule': A key for linking ophiolites, intra-oceanic forearcs, and subduction initiation. Whattam, S. A. & Stern, R. J.] [COPYRIGHT] (2011).

subduction-initiation event, we derive that the minimum amount of oceanic lithosphere subducted beneath the arc was 3560 km, with a likely similar ocean width. However, most tectonic convergence rates are 3–6 cm/year (30–60 km/Ma), so this estimate of ocean width is a minimum. We note that paleomagnetic data from the Pilbara craton suggests that Archean plate velocities were likely similar to those of the modern Earth[17], so our conservative estimates are reasonable. Thus, the minimum horizontal transport distance (relative to the subducting plates) of the nappes is ~3560 km from where the metabasic rocks of the forearc assemblage formed. Further work may reveal if the presently preserved *schistes lustrés* preserve this entire history, though it is likely that the shear zone evolved from the initial oceanic metamorphic sole, was slowly replaced during transport, and the *schistes lustrés* records only the final emplacement of the nappes over the sedimentary sequence on the passive margin of the Eastern Block of the NCC. This is also supported by our observation of a decrease in metamorphic grade from amphibolite facies in nappes to greenschist facies in the underlying *schistes lustrés*.

We document a ~2.7 Ga picrite-boninite-IAT association in subhorizontal nappes in the Archean Central Orogenic Belt of the NCC. The COB includes products of Archean plate convergence, indicating that a once widely separated arc and continent collided and experienced extensive crustal shortening (Fig. 10b, c).

Analogous to classical orogenic belts in the extant plate tectonic regime, the 1600-km-long COB exhibits well-defined tectonic zones (Figs. 1, 10c), including: (1) hinterland of strongly deformed intra-oceanic arc magmatic rocks; (2) circa 2.5 Ga ophiolitic-tectonic mélange; (3) 15–20 km wide belt of ductile fold nappes and thrusts, with forearc-affinity assemblages; (4) a 2.7–2.5 Ga passive-margin sequence, overlain locally by a 2.5 Ga foreland basin, now preserved as several relict sequences including the Qinglong, and Songshan sequences (2.50–2.45 Ga); (5) underlying ancient crystalline basement of the Eastern Block of the NCC. This tectonic zonation is comparable with young orogens, e.g., the Cenozoic Austro-Alpine orogen, grading from highly-deformed and metamorphosed hinterlands, through zones of nappes, to foreland fold-thrust belts, and eventually into relatively undeformed foreland basins overlying passive-margin sediments and underlying crystalline basement[5,6]. Therefore, the multi-dimensional resemblance of (1) geometry and kinematics, (2) contained rock suites, and (3) the tectonic zonation relationship in both ancient and young orogenic belts indicate that tectonic style and the mechanical behavior of lithosphere has had little or no change in rheology, and hence thermal character, since the Archean.

Together, our documentation of the Mariana-type forearc subduction-initiation record, in far-traveled nappes emplaced over a continental margin sequence, represents a solid data point

in time, and a step in understanding the dynamics of subduction through time[2] that modern style, horizontal plate tectonics was operating on Earth in the late Archean, with horizontal plate translations of thousands of kilometers. Thus, proposals that subduction and modern-style tectonics did not initiate on Earth until ~0.80 Ga[2], or 0.52 Ga[49], are incorrect, and the time of the onset of plate tectonics must precede 2.7 Ga[10,57]. Further, our demonstration of subduction initiation, which formed an upper-plate island-arc sequence that was active for ~178 million years, shows that Archean subduction was long-lived, potentially deep[16], and that the Archean Earth had interactions of the crustal and mantle geodynamic systems, similar to those in the present extant mobile-lid regime, at critical periods in the evolution of planetary dynamics, and evolution of life. Our documentation of subduction initiation at ~2.7 Ga shows that the planet's inter-related systems, such as carbon recycling, with broad consequences for the rise of atmospheric oxygen and the development of life on Earth[2], were already in operation by 2.7 Ga.

## Methods

**Geological mapping and structural analysis**. The geological map was constructed using modern methods of litho-structural mapping, in which we subdivided the mapping area into several litho-structural domains based on the map distribution of individual nappes. Mapping was performed at a scale of 1:666. The profiles shown in Figs. 1 and 2 have no vertical exaggeration so represent the true orientation of the folds, and they are oriented sub-perpendicular to the fold axes in the region, on which the structural elements (e.g., foliation, lineation, fold axis, axis plane, thrust fault) were carefully collected, measured and depicted in lower-hemisphere equal angle projections, plotted using the software *Stereonet 11.3* (courtesy of Rick Almendinger, Cornell University). Based on the field mapping and structural analysis, samples were carefully selected in well-constrained structural positions, so that the geochemical and geochronological data can be put clearly into their correct structural context for tectonic interpretations.

**Geochemistry and geochronology**. Major and trace element analyses of whole-rock geochemistry were conducted on X-ray fluorescence spectrometry (XRF) (PANalytical, PW2424, Netherlands), inductively coupled plasma-mass spectrometry (ICP-MS) (Agilent, 7700e, USA), and inductively coupled plasma-atomic emission spectrometry (ICP-AES) (Varian, ICP735-ES, USA) at the ALS Chemex (Guangzhou) Co., Ltd. Before analyses, samples were crushed and powdered to 200-mesh in an agate mill; then the sample powders were dried at 105 °C in an oven.

For major element geochemical analyses, one pulp dried subsample powder was fused with lithium metaborate–lithium tetraborate flux and an oxidant (Lithium Nitrate) in a platinum crucible by furnace heating at 1150 °C. Then, the melting sample was poured into a platinum mould and the resultant sample disk is used for XRF spectrometry analyses. The other pulp subsample for determining a loss-on-ignition was processed as follows: (1) accurately weighed when heating for an hour at 105 °C; (2) heated in a furnace and fused at 1000 °C for 2 h, then cooled to ambient temperature and weighed again. The percentage loss on ignition (LOI) is calculated from the difference in weight from 105 to 1000 °C divided by the weight at 105 °C.

For trace element geochemical analyses, one dried subsample powder was digested with perchloric, nitric, and hydrofluoric acids for each sample. The leached residue was evaporated to dryness, and then dilute hydrochloric acid was added and diluted to volume. Then, the solution was analyzed using an Agilent 7700e ICP-MS and Varian ICP735-ES ICP-AES for ultra-trace to trace level elements. The other subsample was mixed with lithium metaborate/lithium tetraborate flux and fused in a furnace at 1025 °C, then cooled and dissolved in nitric, hydrochloric, and hydrofluoric acids. These solutions were then analyzed by an Agilent 7700e ICP-MS. The relative standard derivations (RSD) are within 5%. The accuracies are better than 5–10% for most trace elements for ultra-trace to trace level element. According to the actual situation of the sample and the digestion effect, the comprehensive value is the final test results shown in Supplementary data 2.

Four large samples were collected (18.1–33.95 kg per sample) and chosen for zircon U-Pb dating. The gravimetric and magnetic techniques were applied for zircon grain separation before clean and transparent zircon grains for U-Pb dating were carefully selected and hand-picked under a binocular microscope, then mounted in epoxy resin and polished down to expose the grain centers.

U-Pb isotope and trace element analyses of these zircons were conducted synchronously by LA-ICP-MS at Wuhan Sample Solution Analytical Technology Co., Ltd with an Agilent 7700e ICP-MS instrument and a Geo-Las Pro laser ablation system. Helium was applied as a carrier gas, which is mixed with Argon, the make-up gas, via a T-connector before entering the ICP. The spot size and frequency of the laser were set to 32 μm and 5 Hz in this study. Zircon 91500, GJ-1,

and glass NIST610 were used as external and internal standards for U-Pb dating and trace element calibration, respectively, with twice every 6 analyses during the whole determination. Each zircon spot analysis incorporated a background acquisition of ~20 s succeeded by 50 s of data acquisition[58]. Concordia diagrams and weighted mean age calculations were made using Isoplot/Exver3[59]. The results are presented in Supplementary data 1.

## Data availability

All data used in this contribution is available in the main text, supplementary files, or in the cited papers. Any questions regarding the data should be directed to the communicating authors.

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

## Acknowledgements

This work was supported by the National Natural Science Foundation of China (Grant Numbers: 41890834, 91755213, 41961144020, 41602234, and 41888101), Chinese Ministry of Education (BP0719022), the Chinese Academy of Sciences (QYZDY-SSWDQC017), the MOST Special Fund (MSF-GPMR02-3), and the Open Fund (GPMR201704) of the State Key Laboratory of Geological Processes and Mineral Resources, China University of Geosciences (Wuhan), and the Fundamental Research Fund (CUGL180406) from the China University of Geosciences, Wuhan.

## Author contributions

T.K. and L.W. conceived the project and obtained funding for the field and analytical expenses. All authors (Y.Z., T.K., L.W., A.P., Y.P., Z.L., C.W., J.W., and H.D.) participated in the fieldwork and analysis, and the laboratory work was led by Y.Z. with Y.P. Z.L., X.L., and C.W, supervised by T.K. and L.W. The manuscript was written by Y.Z., T.K., and L.W., with editing by Y.Z., T.K., L.W., A.P., X.L., Y.P., Z.L., C.W., J.W., and H.D.

## Competing interests

The authors declare no competing interests.
