## [Peer Review File · Nature Communications]

Alpine-style nappes thrust over ancient North China continental margin demonstrate large Archean horizontal plate motionsREVIEWER COMMENTS

Reviewer #1 (Remarks to the Author):

Review Report of:

Archean Alpine-style nappes thrust over continental margin, North China, demonstrate large Archean horizontal plate motions

Yating Zhong, Timothy Kusky, Lu Wang, Ali Polat, Xuanyu Liu, Yaying Peng, Zhikang Luan, Chuanhai Wang, Junpeng Wang, Hao Deng

Review by Ícaro Dias da Silva, Montemor-o-Novo, May 2021

Dear authors,

I have carefully read your manuscript and made some comments in the revised PDF file, attached with this review.

The manuscript is about the Late Archean geological evolution of the North China Craton (NCC) and the similarities between the 2.7-2.5 Ga convergence and modern-day collisional orogens, such as the Alpine belt. This work is a big contribution to our comprehension of the tectonic regimes in Earth's history, pushing the barrier of modern orogens first appearance from previous estimates of 800 Ma to 2.7 Ga. The authors present new structural and tectono-stratigraphic data of the Zanhuang complex of the NCC, combining new field data including 1:666 geological map, geochronological and whole-rock geochemistry of igneous rocks (picrites/boninites, trondhjemites, and tholeiites). Data and interpretations are well integrated in the text. The reader is perfectly aware of the novelty and how is supported by data. All figures are well integrated in the text and are necessary, but I think you need to insert your detail map to have a better 3D look at the major (1st and 2nd order) structures. There are more detailed comments in the revised PDF that I think should be addressed.

As a general comment, I think that the authors underestimate the value of the metamorphic imprint of each tectonic slice of the Zanhuang Complex. I miss some petrographic/microtectonic studies of the relation between fabrics and porphyroblast/porphyroclasts (namely the garnet ones), which I believe are important to understand the tectono-metamorphic evolution of this region, further supporting the authors' suspicions. Likewise, I miss the link between sedimentation in the basins surrounding the orogenic highs (backarc, forearc and foreland), while exploring the possibility of occurrence of polygenetic mélanges in the suture zone between the Eastern and Western NCC blocks, with special incidence in the study area (namely in the MORB tectonic mélange and in the underlying Black Rock and Buddha nappes). This can also be an interesting link with modern orogens that you may explore. I suspect (and expect) that this work is the beginning of much larger study which ultimately will confirm the expectations raised by the authors in this manuscript.

In overall, although the manuscript can be controversial, it is nicely written and has enough novelties supported by data, to be published in Nature Communications, after minor to moderate alterations to its original form.

Thank you for giving me the opportunity to review your work!

Kindest regards,

Ícaro Dias da Silva

Instituto Dom Luiz, Faculdade de Ciências da Universidade de Lisboa

Reviewer #2 (Remarks to the Author):

The manuscript "Archean Alpine-style nappes thrust over continental margin, North China, demonstrate large Archean horizontal plate motions" presents Archean picrites, boninites, and island arc tholeiites (IAT) within Archean-age sub-horizontal nappes. The existence of plate tectonics before the Proterozoic is hotly debated. While there is already published evidence for

subduction in late Archean rocks (e.g Jenner et al., 2009), the combination of geochemical data (picrites, boninites and tholeiites), which indicates a subduction setting, together with clear documentation of sub-horizontal nappes (which indicate accretion), presents excellent and rare evidence for an example of horizontal, modern-style tectonics in the Archean. The example presented here is remarkably similar to modern/non-Archean orogens, and this work is novel and important for advancing our understanding of early earth dynamics.

The methods used in the work are sound. The field documentation and mapping are detailed and of high quality, and the geochemical methods follow current standards. The interpretations presented in the work are reasonable and convincing based on the data. The geochemistry plots and age dating are also reproducible based on the supplementary information. My main concern with the manuscript is that there is too much important information that has been presented as supplementary, instead of within the manuscript. In particular, the ages presented as part of supplementary data (Geochronology of the Zanhuang fold nappe units) are key to the interpretations of the work (e.g quantifications of Archean plate motions), and I therefore recommend that they should be incorporated as part of the main manuscript. In addition, given that a significant focus of the manuscript is on the igneous geochemistry of metamorphic rocks, the question of element mobility (due to metamorphism) should also be at least briefly addressed within the manuscript rather than only within the supplementary information.

Overall, the findings in this manuscript are interesting and important for the advancement of our understanding of early plate tectonics. The work is novel and of significant interest to the community.

Most of my comments can be found in the attached pdf of the manuscript. Below are more general comments:

Lines 126-167

- The authors present stereonet for two phases of deformation (D1 and D2) and mention that two phases of deformation pre-date granite intrusion at c. 2.5 Ga, however the relationship between these two deformation events is not described or discussed. How do each of these two events manifest in the field? What is the difference between them? Why is D1 only in the Black Rock Temple Nappe? Is the foliation and folding described in lines 139-147 related to D1 or D2? Which deformation phase are the microstructures in the rocks (e.g. snowball garnets, asymmetric garnet porphyroblasts) related to?
- There is little mention of metamorphic grade in this section and in general. Since element mobility during deformation and metamorphism is addressed in a supplement, it is clear that the authors are aware that metamorphic overprinting can be a problem for the geochemical data. In my opinion the metamorphic grade of the rocks in each nappe should be more clearly presented in the manuscript and not just mentioned in a supplement.

Lines 235-249

- In the manuscript the evidence for the schistes lustres layer being a metapelite is not clear. Early in the manuscript it is described as a metapelite, and then later as having high Mg, Fe and low Ca content. A metapelite typically has a high Al content. In this section the description of high proportions of chlorite and amphibole (and lack of Al-bearing minerals) also indicates that this is not a metapelite. There is no XRF analysis of the schistes lustres presented in the manuscript or supplementary information to check. I am confident that the schistes lustres is a metasedimentary rock, based on the presentation of detrital ages in the supplement (Geochronology of the Zanhuang fold nappe units), but I am not convinced it is a metapelite.
- The chlorite in the schistes lustres layer suggests that the shearing and metamorphism along this layer is at a lower metamorphic grade than the amphibolite facies metamorphism in the nappes (I assume based on the garnet amphibolites). Assuming the amphibolite facies metamorphism in the nappes described in the manuscript was as a result of nappe emplacement (I think your metamorphic ages support this), you could use the observation of a decrease in metamorphic grade in the schistes lustres to support your interpretation in line 501.

Lines 245-249

- The point made here on the timing of nappe-emplacment is important and needs to be emphasized more as it is the basis of the estimate of the time period (178 Ma) between subduction initiation and nappe-emplacment. Later it is also stated that emplacment took place at 2520 Ma, which is based on the maximum depositional age of the schistes lustrés. However, based on the error on the age estimate, and the intrusion of the 2.5 Ga granite, the emplacment age could be anywhere between 2520 -2500 Ma and this needs to be made clearer (e.g. in lines 481-482) as this affects the time between subduction initiation and nappe-emplacment, and therefore the estimate for amount of subducted oceanic crust.

Supplementary information – Assessment of element mobility

- In the supplement the authors state that a low LOI value indicates no significant hydration of the rocks. In my opinion, the presence of hydrous metamorphic minerals amphibole and various micas indicate the mafic rocks have been significantly hydrated during metamorphism.

Supplementary information – Geochronology of the Zhanhuang fold nappe units

- It should be made clearer for the reader which nappes each of the geochron samples come from.
- Line 17 – cathodoluminescence is misspelled.

References

Jenner, F.E., Bennett V.C., Nutman, A.P., Friend, C.R.L., Norman, M., Yaxley, G. 2009. Evidence for subduction at 3.8 Ga: geochemistry of arc-like metabasalts from the southern edge of the Isua Supracrustal Belt. *Chem. Geol.* 261, 83–98.

REVIEWER COMMENTS

Reviewer #1 (Remarks to the Author):

Review Report of:

Archean Alpine-style nappes thrust over continental margin, North China, demonstrate large Archean horizontal plate motions

Yating Zhong, Timothy Kusky, Lu Wang, Ali Polat, Xuanyu Liu, Yaying Peng, Zhikang Luan, Chuanhai Wang, Junpeng Wang, Hao Deng

Review by Ícaro Dias da Silva, Montemor-o-Novo, May 2021

Dear authors,

I have carefully read your manuscript and made some comments in the revised PDF file, attached with this review.

The manuscript is about the Late Archean geological evolution of the North China Craton (NCC) and the similarities between the 2.7-2.5 Ga convergence and modern-day collisional orogens, such as the Alpine belt. This work is a big contribution to our comprehension of the tectonic regimes in Earth's history, pushing the barrier of modern orogens first appearance from previous estimates of 800 Ma to 2.7 Ga. The authors present new structural and tectono-stratigraphic data of the Zhanhuang complex of the NCC, combining new field data including 1:666 geological map, geochronological and whole-rock geochemistry of igneous rocks (picrites/boninites, trondhjemites, and tholeiites). Data and interpretations are well integrated in the text. The reader is perfectly aware of the novelty and how is supported by data. All figures are well integrated in the text and are necessary, but I think you need to insert your detail map to have a better 3D look at the major (1st and 2nd order)

structures. There are more detailed comments in the revised PDF that I think should be addressed.

Thank you for the great comments. We have gone through every comment in the annotated pdf, and made a separate file showing how we have considered, and addressed each comment, incorporating those that we can. Your comments are very insightful, and relate to on-going work- this paper is a discovery paper, and we are actively working in the area on many of the points that you suggest, but it will take us about 1-2 more years to finish the metamorphic and geometrical details that the reviewer mentions. We have answered all of the detailed comments from the pdf file sent by Ícaro Dias da Silva and the second reviewer, and list the specific replies in a separate file.

As a general comment, I think that the authors underestimate the value of the metamorphic imprint of each tectonic slice of the Zhanhuang Complex. I miss some petrographic/microtectonic

studies of the relation between fabrics and porphyroblast/porphyroclasts (namely the garnet ones), which I believe are important to understand the tectono-metamorphic evolution of this region, further supporting the authors' suspicions.

Thank you, astute comment! This is exactly the topic of first author's PhD work. It is complex, involving lots of analytical work. From what we know at present, the metamorphic history is compatible with the conclusions of this manuscript, which is based on the main mapping, profile construction, geochronology, geochemistry, and detrital zircon work. So, yes, we agree, but this short paper is focused on the discovery of the nappes, the shear zone emplacing them over the passive margin, and the documentation that the nappes contain a forearc subduction initiation sequence. The age difference from the subduction initiation, to the emplacement, gives us the time duration of subduction, and using a very conservative 2 cm/yr rate, we get a minimum horizontal plate transportation distance for the Archean.

The geometry, and metamorphic grades in each nappe, are quite interesting as well, but not essential for the general conclusion above. We will return to that in future work.

Likewise, I miss the link between sedimentation in the basins surrounding the orogenic highs (backarc, forearc and foreland), while exploring the possibility of occurrence of polygenetic mélanges in the suture zone between the Eastern and Western NCC blocks, with special incidence in the study area (namely in the MORB tectonic mélange and in the underlying Black Rock and Buddha nappes). This can also be an interesting link with modern orogens that you may explore. I suspect (and expect) that this work is the beginning of much larger study which ultimately will confirm the expectations raised by the authors in this manuscript.

You are correct. We are proposing to start a major project based on these discoveries, testing exactly what you suggest. But, like you note, this is ongoing work, it will take us time to obtain new funding and form an interdisciplinary team for the new project. We totally agree with these questions, we need go to the field with various expert teams, which we are proposing to funding agencies.

In overall, although the manuscript can be controversial, it is nicely written and has enough novelties supported by data, to be published in Nature Communications, after minor to moderate alterations to its original form.

Thank you!!

Thank you for giving me the opportunity to review your work!
Kindest regards,

Thank you for such a positive and constructive review!

Ícaro Dias da Silva
Instituto Dom Luiz, Faculdade de Ciências da Universidade de Lisboa

Reviewer #2 (Remarks to the Author):

The manuscript “Archean Alpine-style nappes thrust over continental margin, North China, demonstrate large Archean horizontal plate motions” presents Archean picrites, boninites, and island arc tholeiites (IAT) within Archean-age sub-horizontal nappes. The existence of plate tectonics before the Proterozoic is hotly debated. While there is already published evidence for subduction in late Archean rocks (e.g., Jenner et al., 2009), the combination of geochemical data (picrites, boninites and tholeiites), which indicates a subduction setting, together with clear documentation of sub-horizontal nappes (which indicate accretion), presents excellent and rare evidence for an example of horizontal, modern-style tectonics in the Archean. The example presented here is remarkably similar to modern/non-Archean orogens, and this work is novel and important for advancing our understanding of early earth dynamics.

Thank you for such positive comments. We do our best to demonstrate an example of Phanerozoic-style plate tectonics on the Archean earth by adopting comprehensive geologic research based on the main mapping, profile construction, geochronology, geochemistry, and detrital zircon work. Such a multi-disciplinary-methodology allows us to advance our knowledge, leading to and supporting the conclusions of the paper.

The methods used in the work are sound. The field documentation and mapping are detailed and of high quality, and the geochemical methods follow current standards. The interpretations presented in the work are reasonable and convincing based on the data. The geochemistry plots and age dating are also reproducible based on the supplementary information. My main concern with the manuscript is that there is too much important information that has been presented as supplementary, instead of within the manuscript. In particular, the ages presented as part of supplementary data (Geochronology of the Zanhuang fold nappe units) are key to the interpretations of the work (e.g. quantifications of Archean plate motions), and I therefore recommend that they should be incorporated as part of the main manuscript. In addition, given that a significant focus of the manuscript is on the igneous geochemistry of metamorphic rocks, the question of element mobility (due to metamorphism) should also be at least briefly addressed within the manuscript rather than only within the supplementary information.

Thank you for your valuable suggestion, we now mention the assessment of element mobility more in the text, and cover that in detail in the supplementary data. Because of space limitations,

we can only discuss the final ages with errors in the main text, and present the details of the geochronology in the supplementary data.

Overall, the findings in this manuscript are interesting and important for the advancement of our understanding of early plate tectonics. The work is novel and of significant interest to the community.

Thank you for such positive comments.

Most of my comments can be found in the attached pdf of the manuscript. Below are more general comments:

We have addressed all of the comments from the attached pdf in the separate response to reviewers' letter.

Lines 126-167

- The authors present stereonet for two phases of deformation (D1 and D2) and mention that two phases of deformation pre-date granite intrusion at c. 2.5 Ga, however the relationship between these two deformation events is not described or discussed. How do each of these two events manifest in the field? What is the difference between them? Why is D1 only in the Black Rock Temple Nappe? Is the foliation and folding described in lines 139-147 related to D1 or D2? Which deformation phase are the microstructures in the rocks (e.g., snowball garnets, asymmetric garnet porphyroblasts) related to?

D1 structures include mesoscopic fold (F1) and associated planar fabrics (S1). F1 are rarely preserved and limited to several decimeter-scale folds mainly composed of tight-isoclinal, rootless intrafolial folds. The related foliation S1, overprinted by later slaty-cleavage or gneissic foliation (S2), is important evidence for transposition, indicating a transformation of initial foliation (S1) by folding, ductile shear into parallelism with later planar fabrics (S2) through progressive deformation.

D2 structures are the most predominant structures and are responsible for the essential framework of the fold nappe thrust belt that can be traced for km's through central Zhanhuang massif. The shallow-dipping foliation S2 is ubiquitous, on which a pervasive lineation (L2), characterized by a NW-SE trending lineation and a top-to-the SE shearing. F2 folds are mainly

tight-to-isoclinal recumbent folds and asymmetric folds with thickened hinges and attenuated overturned limbs.

The second deformation (D2) phase is related to the foliation and folding described in lines 143-154, as well as the microstructures in the rocks (e.g., snowball garnets, asymmetric garnet porphyroblasts).

We address these comments in the revised manuscript, and provide detailed answers in the file "specific replies to detailed comments" file.

- There is little mention of metamorphic grade in this section and in general. Since element mobility during deformation and metamorphism is addressed in a supplement, it is clear that the authors are aware that metamorphic overprinting can be a problem for the geochemical data. In my opinion the metamorphic grade of the rocks in each nappe should be more clearly presented in the manuscript and not just mentioned in a supplement.

Thanks for your valuable comment! The metamorphism study is exactly the further research being done by the first author. Based on our limited understanding so far, the metamorphic grade in this section and in general are amphibolite to upper amphibolite facies, as described in the text and shown in several photomicrographs.

Lines 235-249

- In the manuscript the evidence for the schistes lustrés layer being a metapelite is not clear. Early in the manuscript it is described as a metapelite, and then later as having high Mg, Fe and low Ca content. A metapelite typically has a high Al content. In this section the description of high proportions of chlorite and amphibole (and lack of Al-bearing minerals) also indicates that this is not a metapelite. There is no XRF analysis of the schistes lustrés presented in the manuscript or supplementary information to check. I am confident that the schistes lustrés is a metasedimentary rock, based on the presentation of detrital ages in the supplement (Geochronology of the Zanhuang fold nappe units), but I am not convinced it is a metapelite.

Answer: you are right, if it is a metapelite one would expect high Al. It is a complex shear zone unit with many components, and we have on-going work to advance our knowledge of the petrogenesis of schistes lustrés units, with XRF analysis for sure. So, taking this comment to heart, we now refer to the unit as a dominantly metasedimentary unit as our investigations are on-going. In the Alpine system, it took the researchers more than 100 years to fully characterize the schistes lustrés units to get to the present level of understanding, and we are making the first report of the similar rocks from the Archean in this manuscript.

- The chlorite in the schistes lustrés layer suggests that the shearing and metamorphism along this layer is at a lower metamorphic grade than the amphibolite facies metamorphism in the nappes (I assume based on the garnet amphibolites). Assuming the amphibolite facies metamorphism in the nappes described in the manuscript was as a result of nappe emplacement (I think your metamorphic ages support this), you could use the observation of a decrease in metamorphic grade in the schistes lustrés to support your interpretation in line 501.

Good comments, that helps! We revised the text by adding “This is also supported by our observation of a decrease in metamorphic grade from amphibolite facies in nappes to greenschist facies in underlying schistes lustrés.”

Lines 245-249

- The point made here on the timing of nappe-emplacement is important and needs to be emphasized more as it is the basis of the estimate of the time period (178 Ma) between subduction initiation and nappe-emplacement. Later it is also stated that emplacement took place at 2520 Ma, which is based on the maximum depositional age of the schistes lustrés. However, based on the error on the age estimate, and the intrusion of the 2.5 Ga granite, the emplacement age could be anywhere between 2520 -2500 Ma and this needs to be made clearer (e.g., in lines 481-482) as this affects the time between subduction initiation and nappe-emplacement, and therefore the estimate for amount of subducted oceanic crust.

Good comments! Here we used maximum depositional age (2520Ma) of detrital zircon in schistes lustrés, which constrain the oldest age of the nappes emplacement and provide a minimum value (178Ma) to the period from birth to death accordingly. We do indeed discuss the additional constraint of the cross-cutting granite dike at 2500 (20 Ma younger), but take the conservative estimate of the collision and lateral transport ending by 2520, with the thermal and igneous effects of the collision continuing until ~2500 Ma.

Supplementary information – Assessment of element mobility

- In the supplement the authors state that a low LOI value indicates no significant hydration of the rocks. In my opinion, the presence of hydrous metamorphic minerals amphibole and various micas indicate the mafic rocks have been significantly hydrated during metamorphism.

Sure, the micas have structural water, but in metapelites, the micas are largely detrital, from the original sedimentation. But once again, this discussion is not directly related to the main point of this paper, which is about the ages and chemistry of the forearc subduction initiation sequence,

and the difference in age related to emplacement over the passive margin.

Supplementary information – Geochronology of the Zanhuang fold nappe units

- It should be made clearer for the reader which nappes each of the geochron samples come from.

Thank you, we make it more clear in the Supplementary information I, line 5-8, by saying “Four samples from the Zanhuang fold nappe/thrust belt were selected for LA-ICP-MS zircon U–Pb dating, including tholeiitic meta-basalts (19NQ-2 and 18Bd-13) from Buddha nappe, a metamorphic felsic dike (19ZY-2) crosscutting picritic-boninitic metabasite from Black Rock Temple nappe, and schistes lustrés (19SL)”.

- Line 17 – cathodoluminescence is misspelled.

Corrected.

Response to comments in annotated pdf from Reviewers 1 and 2, including simple answers, and how we have changed (or not) the text in response.

Comments:

1. Line 54-55: How old? related to which tectono-metamorphic process?

Response: The older gneissic terrane tectonically belongs to the Eastern Zhanhuang Domain (EZD). We have modified the text to say “which includes rocks with ages ranging from 3.8 Ga to 2.5 Ga, with a major crustal growth event at ~2.7 Ga and a crustal reworking event at 2.5 Ga.”

2. Line 57: within error they are the same age.

Response: Yes, but it is an intrusive relationship, and the intruding rock appears slightly younger. But this is expected as the subduction initiation events typically only take about 10 Ma (see for instance Reagan et al 2017 (ref 8 in references))

3. Line 59-63: Can it be a synorogenic basin with "exotic" blocks (olistoliths) from the overriding obducted MORB rocks (sedimentary mélangé), latter tectonized to form a polygenetic mélangé? see: Festa, A., Ogata, K. & Pini, G. A. Polygenetic mélanges: a glimpse on tectonic, sedimentary and diapiric recycling in convergent margins. Journal of the Geological Society **177**, 551-561, doi:10.1144/jgs2019-212 (2020).

Response: Most of the blocks in matrix, and “lenticular highly-mixed thrust-bounded tectonic slices of different rock types, at multiple scales, in many cases bounded by block-in-matrix type mélangé” have been formed by structural attenuation and disruption, so Wang et al. (2013, 2017) and Kusky et al. (2020) classified it as a tectonic mélangé. In a few layers in the underlying passive margin sequence (deformed) there are clear olistoliths, but we have not clearly identified any in the mélangé. This is discussed in the review paper by Kusky et al. 2020 (ESR), and is not the focus of this paper, which is focused on the

more coherent units in this belt, the giant recumbent fold nappes. The nappe units we describe here are thin, laterally continuous sheets, isoclinally folded into recumbent nappes, not olistoliths.

Wang, J. et al. Petrogenesis and geochemistry of circa 2.5 Ga granitoids in the Zanhuang Massif: Implications for magmatic source and Neoproterozoic metamorphism of the North China Craton. Lithos 268, 149–162 (2017).

Wang, J. et al. A late Archean tectonic mélange in the Central Orogenic Belt, North China Craton. Tectonophysics 608, 929–946 (2013).

Kusky, T., Wang, J., Wang, L., Huang, B., & Shi, G. Mélanges through time: life cycle of the world's largest Archean mélange compared with Mesozoic and Paleozoic subduction-accretion-collision mélanges. Earth-Science Reviews, 103303(2020).

4. Line 67: Change Cenozoic to Phanerozoic?

Response: Fixed, thanks.

5. Line 79-81: Of course, it depends on which section of the oceanic was delaminated and obducted onto the passive margin. In Phanerozoic belts, the obducted ophiolitic slices are usually the hyper extended passive margin segment, with transitional continental-oceanic features. Also, if the "MORB Mélange" is a polygenetic (diapiric/sedimentary overlapped by tectonic) mélange, interpretation can be somewhat different, with different estimates of horizontal displacements.

Response: The obducted oceanic fragment is SSZ-affinity picrite-boninite-IAT-MORB basalt association, indicative of subduction initiation magmatism, rather than hyperextended passive margin segment of transitional continental-oceanic features. Therefore, the age constrained from obducted oceanic fragments can be used for the estimation of horizontal displacements. Also as discussed above, and in the papers cited (Wang et al., 2013, 2017; Kusky et al., 2020) it is clearly documented that the associated higher-level mélange is a tectonic ophiolitic mélange. We thus have considered different possible origins for the oceanic (ophiolitic) rocks and have shown clearly that they are fragments of a forearc crust/mantle sequence. This was also described recently from a different location in this orogen by Ning et al. 2020 (Precam. Res). We agree that many Variscan ophiolites are hyperextended continental margins (see, for example reviews in Kusky et al., 2011, or Dilek and Furnes, 2011), but the examples we

are describing are different, and are clear fragments of forearc lithosphere.

[1] Ning, W. *et al.* *From subduction initiation to arc-polarity reversal: Life cycle of an Archean subduction zone from the Zunhua ophiolitic mélange, North China Craton. Precambrian Research*350, 105868 (2020).

[2] Kusky, T. M., Lu, W., Dilek, Y., Robinson, P., Songbai, P., & Xuya, H. *Application of the modern ophiolite concept with special reference to Precambrian ophiolites. Science China Earth Sciences*, 54(003), 315-341 (2011).

[3] Dilek, Y., & Furnes, H. *Ophiolite genesis and global tectonics: geochemical and tectonic fingerprinting of ancient oceanic lithosphere. Geological Society of America Bulletin*, 123(3-4), 387-411 (2011).

6. Line 81-82: Metamorphism in the nappe pile and underlying units? Are there geochronological ages of the thrust zones?

Reply: Yes, the nappe pile and underlying units recorded metamorphism event corresponding to the orogeny. We have not directly obtained geochronological ages of thrust zones, but the age can be constrained by geological crosscutting relationships, e.g., the one reported in Wang *et al.*, 2013 that intrusions of a circa 2.5 Ga granitic pluton and undeformed pegmatites crosscutting mélange and D2 foliation. Therefore, the geochronological ages of the thrust zones should not be younger than circa 2.5 Ga, which is consistent with our estimation.

Wang, J. *et al.* *A late Archean tectonic mélange in the Central Orogenic Belt, North China Craton. Tectonophysics*608, 929-946 (2013).

7. Line 94: Maybe it is a good idea if you associate the deformation with the metamorphism. Is it prograde or retrograde? is there evidence of pre-thrusting metamorphism?

Reply: Thanks for your valuable comment! The study on metamorphism-deformation is exactly the further research of which the senior author is doing at the moment. Based on our limited understanding so far, metamorphic grades in this section and in general are amphibolite to upper amphibolite facies, with an apparent increase in grade higher in the nappe stack, but this requires further documentation. We now mention the major metamorphic jump from the upper nappes at amphibolite facies, to the underlying *schistes lustrés* at greenschist facies.

8. Line 109: is it a synorogenic gneiss dome developed in the passive margin?

Response: It is migmatite of the older continental fragment of the Eastern Block of the NCC. Part of our team is making structural maps of the gneiss terrane, since little work has been done there, but the mapping program will take another two years to complete. It could be a Variscan-style gneiss dome, or just a gneiss complex, but that is not part of the discovery presented in the current work.

9. Line 112-113: Is it only tectonic? It may be polygenetic...

Reply: Thanks for pointing out this possibility, but we have been mapping this *mélange* for many years, and we do not see evidence for a pre-deformation olistostromal (i.e., polygenetic) history; it is only tectonic. According to the comprehensive geologic research summarized by Kusky et al., 2020, *Earth Science Reviews*, the tectonic *mélange* in our study area is just one of several Neoproterozoic *mélanges* exposed along Central Orogenic Belt (COB) of the North China Craton. And there is, so far, no diagnostic evidence of polygenetic features for them (Jianping-Zunhua-Zanhuang-Dengfeng (JZZD) *mélanges*) based on systematic field, structural, petrologic, geochronologic, and geochemical studies. But this is such a valuable question that we will consider it in our further research!

Kusky, T., Wang, J., Wang, L., Huang, B., & Shi, G.. (2020). Mélanges through time: life cycle of the world's largest archean mélanges compared with mesozoic and paleozoic subduction-accretion-collision mélanges. Earth-Science Reviews, 103303.

10. Line 114-115: Late to post orogenic magmatism?

Reply: It is post orogenic magmatism, A-type granite defined by Wang et al., 2013, 2017, as described and cited in the text.

Wang, J. et al. Petrogenesis and geochemistry of circa 2.5 Ga granitoids in the Zanhuang Massif: Implications for magmatic source and Neoproterozoic metamorphism of the North China Craton. Lithos 268, 149–162 (2017).

Wang, J. et al. A late Archean tectonic mélanges in the Central Orogenic Belt, North China Craton. Tectonophysics 608, 929–946 (2013).

11. Fig 2: Q1: These are sub-horizontal... is there vertical flattening overprinting the thrust

fabrics?

Q2: These orthogneisses... which age do they have? can these be exotic blocks or are deformed intrusions (dykes?)/in situ melts?

Q3: are these stretching lineations parallel to nappe and thrusting transport direction? If so, these NNE-SSW lineations are related to what? Lateral escape due to collapse?

Q4: micaschists? paragneisses? do you have U-Pb ages for these meta-sediments?

Q5: It would be nice if you also insert this map. This will give a better view on the overall 3d structure. How large is the area? Have you tried to follow these structures along the Zanhuang complex?

Answer 1: There are large sub-horizontal shear strains, with folds elongated into the regionally uniform stretching direction, as described. Vertical flattening would be associated with oblate strain with stretching in all sub-horizontal directions, which is not present, so we can discount large late vertical flattening strains.

Answer 2: Though we have no data on this topic right now, and it is not directly related to the topic of this current paper, it is one of the issues we will work on later as the research continues.

Answer 3: Yes, the stretching lineations are parallel to the kinematics of nappe transport. The fold hinges are rotated variably into parallelism, in sheath-type manners, related to thrusting, but we see no evidence for orogenic collapse in this region. This is described here in the text.

Answer 4: It is paragneiss, we do not have U-Pb ages for these meta-sediments right now, but we will have in the future.

Answer 5: Our mapping so far is in the form of a cross-strike profile corridor rather than a traditional quadrangle-style geologic map. We are making a proposal for a large mapping and geophysical program to map the Zanhuang Complex, which is large, and will take many years of multidisciplinary studies.

12. Line 122: "local place name", remove "place".

Reply: Done!

13. Line 122: Location of cross sections in Fig 1?

Reply: Yes, the location of the cross section is shown in Fig 1.

14. Line 124 and line 129: can you also present this map? It would be nice if you also insert this map. This will give a better view on the overall 3d structure. How large is the area? Have you tried to follow these structures along the Zanhuan complex?

Reply: It is the profile, we have clarified this point in the text, good point though! Thank you!

15. Line 132-133: This "smells" like a flysch synorogenic basin. What is the metamorphic grade of these rocks? And the cherts are related to what? Shale and greywacke flysch (?) sedimentation environment usually does not allow the precipitation of silica gels (these occur under calm sedimentation conditions), unless they are related to volcanism (e.g., VHMS deposits). In this latter case, silica gels usually combine with iron oxides and pyrite to form volcano-exhalative jaspers. To see an example of synorogenic basins associated to thrust tectonics in the late Paleozoic Variscan belt: *González Clavijo, E. et al. A tectonic carpet of Variscan flysch at the base of a rootless accretionary prism in northwestern Iberia: U–Pb zircon age constrains from sediments and volcanic olistoliths. Solid Earth***12**, 835-867, doi:10.5194/se-12-835-2021 (2021).

Reply: These rocks were initially accreted during the accretionary stage of the orogenesis, where the fore-arc ophiolitic fragments (ophirags, of Sengör and Natal'in, 1992), were accreted along with their Ocean Plate Stratigraphy (OPS) packages. OPS is the sedimentary carapace overlying oceanic crust (see Kusky et al., 2013), and typically grades (above the basalt) from chert and / or limestone, to deep water shales, to siltstones, to graywackes, to conglomerates or olistostromes, formed as the oceanic crust is either entering the trench, or being obducted over a passive margin. There are some detrital zircons associated with thrusting in the schistes lustrés unit, but the syn-orogenic foreland deposits are not well-preserved in this mountain range. We describe them from other ranges in this orogenic belt (e.g., Huang et al., 2020; Kusky and Li, 2003),

[1] Sengör, A.M.C, Natal'in, B.A., *Phanerozoic Analogues of Archaean Oceanic Basement Fragments: Altaid Ophiolites and Ophirags*[J]. *Developments in Precambrian Geology*, 13(04):675–726(2004).

[2] Kusky, T. M., Windley, B. F., Safonova, I., Wakita, K., Wakabayashi, J., & Polat, A., et al. *Recognition of ocean plate stratigraphy in accretionary*

orogens through earth history: a record of 3.8 billion years of sea floor spreading, subduction, and accretion. Gondwana Research, 24(2), 501-547 (2013).

[3] Huang, B., Kusky, T. M., Johnson, T. E., Wilde, S. A. & Fu, D. Paired metamorphism in the Neoproterozoic: A record of accretionary-to-collisional orogenesis in the North China Craton. *Earth and Planetary Science Letters* 543, 116355 (2020).

[4] Kusky, T. M., & J Li. Paleoproterozoic tectonic evolution of the North China Craton. *Journal of Asian earth sciences, 22(4), p. 383-397(2003).*

16. Line 134: are these syn, late or post orogenic/kinematic intrusions?

Reply: These are post orogenic/kinematic intrusions, defined by Wang et al., 2013, 2017, as cited in the text.

[1] Wang, J. et al. Petrogenesis and geochemistry of circa 2.5 Ga granitoids in the Zanhuang Massif: Implications for magmatic source and Neoproterozoic metamorphism of the North China Craton. *Lithos* 268, 149-162 (2017).

[2] Wang, J. et al. A late Archean tectonic mélange in the Central Orogenic Belt, North China Craton. *Tectonophysics* 608, 929-946 (2013).

17. Line 134-135 and Line 140-141: language improvement (1) change "is" to "are"; (2) change "imbrications of thrust sheets and nappe generation" to "an imbricated thrust and nappe complex"; (3) remove "are predominant at all scales"; (4) change "(15 - 40°)" to "15 - 40°"; (5) remove "towards the NW".

Line 134-135 and Line 140-141: Improved, thanks.

18. Line 153: metamorphic grade for picrite-boninite-series volcanics and island arc tholeiite basalts (IAT)

Reply: It is amphibolite facies, and we now state this more clearly in the text on lines 162, 218, 236 and 248. However, the metamorphic history of the nappes is very complex and will take years of work to decipher. Though there is no quantitative P-T condition for it so far, it is amphibolite facies by rough estimate based on peak metamorphic mineral association that picrite-boninite-series volcanics are composed of amphibole and plagioclase, IATs are comprised of amphibolite, plagioclase, garnet. The metamorphic assemblages are illustrated in photomicrographs in Fig 5.

19. Line 157-158: is "sheared mica-quartz-rich metapelites (schists lustrés)" the phyllonite?

An: It is a phyllonite. Thank you, good point! We now mention this in the text, lines 167-179 in text

20. Line 160: "Nappes of the entire domain are detached from the deeper substrate and thrust over the autochthonous sequence of marble-siliciclastic passive continental margin strata." You want to know the metamorphic grade and whether it is in prograde or retrograde of "the deeper substrate".

Reply: This is an excellent question! It is related to on-going work. Our work so far suggests that the metamorphic grade is higher in higher nappe stacks, for instance, on lines 165-167 and 533-535 we now note that the amphibolite grade of the Budda nappe rests upon greenschist (lower grade) rocks of the schistes lustrés and underlying passive margin. However, the detailed relationships of the contrasting metamorphic PTt paths will take much more analysis to further document, so please wait a couple of years for that project to be completed.

We have changed the text to read "Amphibolite-grade nappes of the entire domain are detached from the deeper substrate and thrust over the lower-grade greenschist facies autochthonous sequence of marble-siliciclastic passive continental margin strata"

21. Line 163, 165: is clastic "shiny" tectonite the phyllonite? Is the another highly schistose thrust upon buddha nappe of phyllitic nature as the "schists lustrés"

Reply: Yes, the main thrusts are characterized by the "shiny" tectonites, for which we adopt the name "schistes lustrés" based on Alpine examples. They can also be classified as tectonites, as we now mention more clearly in the text on lines 167-169. While we mention this classification now, we feel the words phyllonite and tectonite may not be familiar to many readers of Nature Communications, so we mention this terminology here, but retain the more general terms elsewhere in the text.

22. fig 3c: It does not look like a sheath fold. It looks like fold interference, highlighted by outcrop irregular surface. This should be confirmed in a 3d scheme of this exposure. Do you

have an orthogonal view to make this 3d interpretation?

Reply: This is a true sheath fold. Based on our structural analysis and equal area stereoplots, it is characterized by curvilinear fold hinges as shown in Fig 2 that define a well-developed fold hinge girdle with more than 180° of hinge-line curvature, plunging toward the NW, depicting a regional scale fold shape typical of sheath geometry, whilst the elliptical cross sections (Fig. 3c) only occur orthogonal to the shearing direction. In addition, the unified shear sense of kinematic indicators and lack of two sets of axial planes at high angles to each other rule out the interference pattern for the sheath fold form. Therefore, it is evident that these structures are developed by sheath folding during the course of progressive non-coaxial, top- to- SE, shearing.

23. fig 3d: be careful with the kinematics of the snowball garnets. They can give a reverse kinematics if they are deltas or if the porphyroblast grows with rotation within the foliation (syn-tectonic). See chapter 7 of: *Passchier, C. W. & Trouw, R. A. J. Microtectonics. 2nd edn, 366 (Springer-Verlag, 2005).*

Reply: Thank you, that is very helpful! We changed the language to describe it and took off the arrow in the diagram to avoid such an ambiguity. The rocks indeed have suffered large syn-metamorphic strains and sub-simple shear making the use of rotated porphyroblast systems challenging and ambiguous. As before, we are engaged in systematic structural and metamorphic studies of this region and process, but we still need some more field seasons and work.

24. fig 4d, 4f: Q1: It looks it has post thrusting, vertical flattening...

Q2: This is not axial planar with the nappes. This seems latter re-folding.

Reply 1: According to our regional structural analysis, there seems to be no unambiguous evidence for vertical flattening, as described above in the reply to question 11.

Reply 2: Good point, we now described that the photography is taken from an oblique view of a highly curved axial surfaces and hinge lines in a sheath fold

(see stereonet), thus, the different curvature from the interior of the fold to the enveloping surface gives the illusion of non-axial planar, or refolded structure. Below, we paste a model of the sheath fold, being developed for a future paper.

25. Line 208: MT-LP amphibolite facies?

Reply: it might be, either MT-LP or MT-MP amphibolite facies. As before, this is an interesting point that we are working on, but is not directly related to the main points of this particular paper.

26. Line 219: "tholeiitic metabasalt folded within less competent quartz-biotite gneisses and biotite-rich schists". you want to know the metamorphic grade of them.

Reply: It is amphibolite facies. Based on our thin section observation, the peak metamorphic mineral association of biotite-rich schists is composed of quartz, plagioclase, biotite and garnet. We now mention this in several places in the manuscript-

We appreciate this reviewer's comments, focused largely on the metamorphic history of the nappes. However, this paper is about documenting the forearc petrological, geochemical and age characteristics of the rocks in the nappes, and documenting the age of emplacement of the nappes over the passive margin. That is then used to estimate the lateral transportation of the nappes in the Archean. We are preparing other papers that focus on the very complex metamorphic history of these nappes, but not this one.

27. Line 222-223: you already say this before

Reply: Deleted, thank you.

28. Line 225: what is the metamorphic grade for garnet porphyroblast-bearing foliated amphibolite?

Reply: It is amphibolite facies based on its peak metamorphic mineral association of plagioclase, garnet and amphibole. We mention this in the text on line 247-248 in revised manuscript, by saying “Peak metamorphic mineral assemblages include amphibole, garnet, plagioclase, and quartz and give amphibolite facies metamorphism conditions.”

29. Line 229: “Garnet forms rotated porphyroblasts with sigmoidal and spiral-shaped inclusion trails making spectacular snowball garnet structures” Be careful. See my comments in the photos 5d and 5e.

Reply: Good comment! We agree with your point on the fact that the shear sense inferred from rotated garnet porphyroblasts can be very complicated. We thus took off the kinematic arrow and just showed the syntectonic snowball garnet. We have derived the kinematic information from other larger scale features.

30. Line 233: what does “metamorphic ages of 2453 ± 45 Ma and 1852 ± 29 Ma” mean?

Reply: metamorphic age of 2453 ± 45 Ma represent the syntectonic zircon recrystallization during D2; metamorphic age of 1852 ± 29 Ma represent the later tectonothermal event. We now say this more clearly in the text on lines 245-247.

31. Line 236-239: indicating which metamorphic grade? is there retrograde metamorphism overprinting it?

Reply: it indicates the greenschist facies to low amphibolite facies. It is related to the syn-tectonic metamorphism, and there is no significant retrograde metamorphism overprinting it.

We say this more clearly in the text now, but we must stick to the main point of the paper, as described above. When we write the metamorphic history of these nappes, it will be a 30 page long paper, and while interesting and important, the metamorphic evolution of the belt is not the main point of this paper.

32. Line 243: Are you sure this is a detrital rock? can it be a felsic/intermediate tuff? If so the "MDA" is the volcanic age, and the other zircons can be xenomorphic or inherited components from the melting basement or host rocks. If it is a detrital rock, it can be part of a synorogenic basin, trapped in the accretionary wedge and thrust onto the passive margin. But I am not sure these are metasediments.

Reply: yes, it is a detrital rock. We show this with the CL images of detrital zircons in the supplementary data, and description of the rock.

Zircon grains in the schistes lustrés sample show an isometric or stubby crystal habit and are dominated by relict cores with rounded to stubby shape and oscillatory zoning with overgrowth rims that vary in diameter and interior texture. The rounded terminations with internal oscillatory zoning of relict cores and their wide range of concordant $^{207}\text{Pb}/^{206}\text{Pb}$ ages spectrum reflect the detrital origin.

33. Line 244–245: In the best chance, it shows the peak of metamorphism, not the end. It only shows that there is a zircon producing event in this age, with high temperatures allowing zircon crystallization. Deformation can continue well beyond the metamorphic peak.

Reply: Good comment! Deformation, for sure, can continue well beyond the metamorphic peak. Nevertheless, the age constrained by geologic crosscutting relationships (intrusions of a circa 2.5 Ga granitic pluton and undeformed pegmatites crosscutting the mélangé and D2 foliation) are consistent with our interpretation for syn-tectonic deformation age by metamorphic zircons. The metamorphism age of 2455 ± 26 Ma might imply the age, more specifically, after the peak of metamorphism as deformation continued, but the cross-cutting 2.5 Ga granites are not deformed. The metamorphic age can be taken, within error,

approximately as age of syn-tectonic deformation accordingly.

34.fig 5d-e: It seems this gives the opposite shear direction. If the garnet is syntectonic, it had gradually incorporated the foliation of the matrix as it grows. In this case, the straight foliation inside the garnet is the same as outside, and therefore it rotated to the left (NW). If it is pre-tectonic, then the inner foliation (inclusion trails) may be dragged by the SEwards kinematics you show. What is the mineralogy of these inclusion trails? are they in equilibria with the garnet? if so, which metamorphic conditions they show? Are they identical to the one observed in the matrix?

Reply: The mineral inclusions in garnet include pl+qtz+ilm+ru+amp, which is identical to the mineral associations in the matrix. It supported the syntectonic snowball garnet formed in the Fold nappe. I agree with your point on the fact that the shear sense inferred from rotated garnet porphyroblasts can be very complicated. We thus remove the kinematic arrow and just show the syntectonic snowball garnet for the present study, as we have a comprehensive kinematic analysis of many porphyroblast and porphyroblast systems underway.

35. Line 511-513: This is tectonized at the base of the nappe system? It can be also incorporated in higher structural tectonic slices, forming polygenetic mélanges, including olistolithic MORB rocks and granitic blocks... This can be important, as this is also a feature of Phanerozoic belts, as observed in the Variscan belt in NW Iberia and other European sections.

Reply: Those are interesting relationships documented from the Variscides, but here, we are just describing what we have observed that this schistose unit is the strongly sheared unit at the base of the nappe. We do not want to speculate on what it might be like if it looked like the Variscides.

36. fig 10: The red arrows indicate the presence of mélanges along the confrontation zone. This is also a very typical aspect of modern orogens.

Reply: that is a good comment! The mélanges along the confrontation zone is indeed a very typical aspect of modern orogens. But the topic of mélanges is addressed in some of our previous papers (*Wang et al., 2017, Kusky et al., 2020*), and this paper is about the nappes with forearc volcanics being thrust as coherent units over a passive margin sequence in the Archean.

[1] Wang, J., Kusky, T., Wang, L., Polat, A., Deng, H., & Wang, C., et al. *Structural Relationships along a Neoproterozoic Arc-Continent Collision Zone, North China Craton. GSA Bulletin* 129, 59-75 (2017)

[2] Kusky, T., Wang, J., Wang, L., Huang, B., & Shi, G.. (2020). *Mélanges through time: life cycle of the world's largest archean mélange compared with mesozoic and paleozoic subduction-accretion-collision mélanges. Earth-Science Reviews*, 103303.

All comments have been revised, including the spelling errors and grammar.

We thank Reviewer 1 for very thoughtful and constructive comments, and thoughts and suggestions for future work!

Response to Comments from Reviewer 2

1. Line 26-27: The methods for determining these ages should be clarified here in the abstract.

Reply: Revised, thank you!

2. Line 29, 53: language improvement

An: Fixed.

3. Line 42-44: Sentence sounds better as: Documentation of geological relationships that unambiguously demonstrate large horizontal motions of plates on Earth in the Archean is critical to test if plate tectonics operated in early times.

Reply: Fixed.

4. Line 64-65: you should add reference for "forearc-ophiolitic remnants".

Reply: thank you, we fixed that by adding reference *Kusky, T., Wang, J., Wang, L., Huang, B., & Shi, G. Mélanges through time: life cycle of the world's largest archean mélange compared with mesozoic and paleozoic subduction-accretion-collision mélanges. Earth-Science Reviews*, 103303 (2020).

5. Fig.1: In many cases it is impossible to see the difference between bedding and foliation symbols on the map, even enlarged to 150%. These need to be larger and/or clearer.

Reply: thank you, good comment. We fixed that by assigning different color,

i. e., brown for bedding and black for foliation.

6. Line 103: Remove "rock". Redundant with use of "lithologies".

Reply: Done!

7. Fig. 2: Stereonets are very small and hard to read. I recommend enlarging them. Line widths for sample and figure labels are also very thin. If this is a full-page figure the line widths might be OK.

Reply: Thank you, good comment. We enlarge the stereonet and increase the line widths for sample and figure labels. It would indeed be nice if this is a full-page figure, perhaps the production team can do that if the paper is accepted.

8. Line 132: "The nappes include units of meta-sedimentary (shale, greywacke and chert) and meta-volcanic origin," What is the metamorphic grade of these rocks?

Reply: It is amphibolite facies. Based on our thin section observation, the peak metamorphic mineral association of biotite-rich schists (shale) is composed of quartz, plagioclase, biotite and garnet. This is now described in the text.

9. Line 134-139: language improvement

Reply: Fixed.

10. Line 140-141: "Foliations formed during folding of the nappes dip shallowly (15 - 40°) NW towards the hinterland/ancient sea, 140 towards the NW (Fig. 2, stereographic projections)." D1 or D2? Folding of nappes or nappe-thrusting? In the NW part of the area? If this is dip direction then it has already been said in the sentence.

Reply: We have improved the language to make this clear.

11. Line 140-143: language improvement

Reply: Thanks. We changed the sentences to "The nappes are tight-to-isoclinal fold structures, overturned with SE-vergence towards the foreland, with their axial surfaces subparallel to the overturned limbs (Figs. 2, 4a-b, d-e)".

11. Line 143:” which are typically the sites of thrust faults” .The meaning here is ambiguous. Is it the tight to isoclinal fold structures in general, or specifically the axial surfaces or the overturned limbs that are the sites of thrust structures?

Reply: the fold structures in Deformation 2 are, in general, tight to isoclinal folds. And the thrust structures always occur at the sites of overturned limbs.

We make this more-clear by re-wording the text on lines 145-148: The nappes are tight-to-isoclinal fold structures, overturned with SE-vergence towards the foreland, with their axial surfaces subparallel to the overturned limbs (Figs. 2, 4a-b, d-e). The overturned limbs are the sites of thrust faults, where individual units are repeated, forming thrust duplex structures (Fig. 3a).

12. Line 145: change “sheath-fold forms” to “sheath folds”

Reply: Thank you, done!

13. Line 151: What grade of metamorphic overprint?

Reply: This is correlated with the period of deformation 3, which is characterized by the open, upright folds and crenulation cleavages. The grade of metamorphic overprint might be greenschist facies based on the D3 mineral association of biotite, muscovite and quartz in metapelite rocks. This metamorphic overprint might be related to the metamorphism reported by *Xiao et al., 2021* that later metamorphic overprint documented in the paragneiss and marble in the Zanhuang complex. The metamorphic grade is now mentioned at several places in the text, and described in a related paper, below, which we cite (number 25 in reference list)

Xiao, D. et al. Neoproterozoic to Paleoproterozoic tectonothermal evolution of the North China Craton: Constraints from geological mapping and Th-U-Pb geochronology of zircon, titanite and monazite in Zanhuang Massif. Precambrian Research 359, 106214, doi:10.1016/j.precamres.2021.106214 (2021).

14. Line 154-158: Shorten sentence to make points easier to follow.

Reply: thank you, we rephrase it as “The lowermost allochthonous nappe is named “Buddha nappe” (after a nearby village name). It is a large-scale, recumbent fold, the overturned limb of which is ~100–150 m thick and highly attenuated due to stretching. The Buddha nappe is resting upon a clean-cut thrust fault (Figs. 1, 2) decorated by highly schistose and sheared mica-quartz-rich metapelites, ranging from 3 cm – 3 m thick.”

15. Line 162-1: Specify which figures.

Reply: Revised. The figures related are Figs. 2, 3d-f, 4f-g.

16. Line 162-2: Is there data to show this? XRF analysis that could be included? If it's a metapelite one would expect high Al.

Reply: you are right, if it is a metapelite one would expect high Al. We have done XRF analyses, and it is high Al, and high Mg (a mafic metasediment?), but we are currently making a thorough study of the variability and composition of the schistes lustrés in various locations, a topic of a different study. Nevertheless, it is the on-going work for us advance our knowledge of the petrogenesis of schistes lustrés, with XRF analysis for sure. For the present study, whose aim is more focused, we feel it is appropriate for us to refer to it as metapelite-dominated unit, or just a metasediment right now, based on the petrography, field characteristics, and nature of the detrital zircons contained in the unit, as we document in the paper. We have modified the text in several places to show the unit is more diverse than simply a metapelite.

17. Line 163: I could not find this in the references.

Reply: Yes, it is a manuscript currently being prepared, about the petrogenesis of schistes lustrés, and is not published yet. Deleted.

18. Line 165: Additional overlying nappes? Which nappes?

Reply: Good comment! We removed “then additional nappes” to clean up the ambiguousness and clarified the text as “The top of Buddha nappe is marked by another highly schistose thrust, which in turn overthrust by another large nappe (Black Rock Temple nappe) with similar dimensions to Buddha nappe, but containing picritic-boninitic lavas described below” .

19. Line 183: Within which lithology?

Reply: The Black Rock Temple nappe is mainly comprised of units of picrite-boninite metabasites within a metasedimentary (greywacke and shale protolith) dominated sequence.

This is re-worded on lines 188-190.

20. Line 196-200: Rephrase. Run on sentence.

Original long sentence now broken into two.

21. Line 203-204: change “greywacke and shale” to “greywacke and shale protoliths”; spelling mistake for “an undeformed”.

Reply: Fixed.

22. Line 211: Other minerals aren't abbreviated. Change for consistency.

Reply: Fixed.

23. Line 213: delete /or

Reply: Fixed.

24. Line 222-224: Repetition from above.

Reply: Deleted.

25. Line 226-228: Why the mineral abbreviations here when not above?

Reply: Fixed.

26. Line 231: D1 or D2?

Reply: D2. Information added in the revised manuscript.

27. Line 233: Could this metamorphic age be related to the above syntectonic crystallization during shearing? This suggests this might be the emplacement age. I see how that is a bit of a problem with the 2.5 Ga granite, though.

Reply: Good comment! The metamorphic age can be taken as approximate age of syntectonic crystallization soon after shearing, which is constrained by the 2520 Ma maximum depositional age (i.e., its younger than that) and the 2500 Ma cross cutting dike (its older than that). Theoretically, metamorphic zircon can form at the peak of metamorphism, and can grow also beyond metamorphic pressure peak in a clockwise path; considering the error range given by U-Pb age, the metamorphism age of 2455 ± 26 Ma might imply the age, a little bit, after the peak of metamorphism, as the rocks slowly cooled.

[1] Chen, R. X., & Zheng, Y. F. *Metamorphic zirconology of continental subduction zones. Journal of Asian Earth Sciences*, 145, 149–176 (2017).

[2] Taylor, R., Kirkland, C. L., & Clark, C. *Accessories after the facts: constraining the timing, duration and conditions of high-temperature metamorphic processes. Lithos*, 239–257 (2016).

28. Line 237: for the unpublished electron probe micro-analysis data, perhaps you can include in supplement.

Reply: Thanks. We have deleted the reference to our on-going electron probe studies, but that does not change the petrographically determined abundances noted here.

29. Line 241: Garnet is not described as part of the mineralogy of the schistes lustrés above?

Reply: In the text we clarified that rotated garnet porphyroblasts are in the mylonitic quartzite associated with the contact of the in schistes lustrés unit

and the metabasite units.

30. Line 244: Interesting that this is the same metamorphic age as the tholeiitic basalt.

Reply: Thank you. We noticed that, too. Very consistent! Seems the data is good.

31. Line 245-249: This is very important to the interpretations of the paper and needs to be emphasized more for the discussion of timing of nappe emplacement and how the 178 Ma time period is defined. "Other data" needs a reference.

Reply: Thank you.

We use the age of subduction initiation based on the age of the boninite-tholeiitic and cross cutting trondhjemitic dike series, and the age of thrusting based on detrital zircons, and the age of a cross cutting granite dike—The metamorphism seems to continue for 20 Ma after the best constraints on the age of thrusting. We make it clearer in the text on lines 267-272, and 504-511 (discussion section) that the 178 Ma time period is defined by "The subduction initiation age is inferred by the crystallization age of forearc affinity metabasalt; whereas the emplacement age of nappes is constrained by the maximum depositional age (2520Ma) of detrital zircon in schistes lustrés, which represents the oldest age of the nappes emplacement and provide a minimum value (178Ma) to the period from birth to death, accordingly."

We have added references for "other data" that has been published before.

[1] Li, J., & Kusky, T. A late archaic foreland fold and thrust belt in the North China Craton: implications for early collisional tectonics. *Gondwana Research*, 12(1-2), 47-66(2007).

[2] Kusky, T. M., & J Li. Paleoproterozoic tectonic evolution of the North China Craton. *Journal of Asian earth sciences*, 22(4), p. 383-397(2003).

[3] Kröner, A., Cui, W.Y., Wang, C.Q., Nemchin, A. A. Single zircon ages from high-grade rocks of the Jianping Complex, Liaoning Province, NE China. *Journal of Asian Earth Sciences* 16, 519 - 532(1998).

32. Line 257: No garnet labeled in photomicrograph.

Reply: thank you. We should clarify that the garnet is present in tholeiitic basalt but shown in fig. 5d-e rather than fig. 5d-e. we changed the sentence to “(c) tholeiitic basalt is strongly foliated and chiefly composed of garnet (shown in d-e), biotite, amphibole, plagioclase and quartz;”

33. Line 292 and Line 306: change “FAM diagram” to “AFM diagram”

Reply: Done!

34. Line 327:and mantle wedge condition?

Reply: The mantle wedge condition refers to the fertile, or refractory from prior melting events. This is explained in the next sentence “Their composition reflects the P-T melting conditions, mantle wedge condition prior to melting (i.e., whether fertile, or refractory from prior melting events), variable contributions of volatiles from the slab, and the aggregation of melt fractions from the source peridotites.”

35. Line 359-360: you think “the mid-ocean ridge basalt (MORB)” should not be capitalized.

Reply: We are using the style published in Nature. It seems it should not be capitalized, e.g. “An advantage of studying mid-ocean-ridge basalts (MORBs) rather than sub-aerially erupted lavas is that the higher eruption pressures beneath the ocean inhibit degassing of volatile elements.” In *Saal, A. E., Hauri, E. H., Langmuir, C. H., & Perfit, M. R. Vapour undersaturation in primitive mid-ocean-ridge basalt and the volatile content of earth's upper mantle. Nature, 419(6906), 451-5 (2002).*

36. Line 360 and Line 369: spelling mistakes.

Reply: Fixed.

37. Line 382: change “solid blue-red lines” to “solid blue and red lines”

Reply: Done!

38. Fig. 8: Q1: All other figures follow horizontal labeling, whereas this figure follows vertical labeling (b below a instead of beside a). Fix for consistency.

Reply: Thank you. Done!

Q2: Melt curve labeling is too small to read.

Reply: Fixed.

39. Line 392: Unnecessary? These are already labeled in the figure.

Reply: Thank you, deleted.

40. Line 432: This acronym (SSZ) has not been previously defined.

Reply: “(supra-subduction zone)” is added after SSZ, Thank you.

41. Line 472: you should change “units of nappes” to “nappe units”.

Reply: Done!

42. Line 482: Where this estimate comes from needs to be better defined in the discussion and supported in the manuscript with age dates. The constraints indicate emplacement was between 2520 and 2500 Ma.

Reply: Here we used maximum depositional age (2520Ma) of detrital zircon in schistes lustrés, the age of cross-cutting granitic dikes (2500 Ma), and the maximum depositional age of the top of the passive margin sequence, to constrain the age of the nappes emplacement between 2520 and 2500 Ma. We have added new text on lines 267–272 to explain this more clearly.

43. Line 492: Is this a reasonable estimate for the Archean? This should perhaps be specified.

Reply: The velocity of plate motion varies in the modern earth, ranging from 1 to 20 cm/yr. so we used conservative estimation for velocity 2 cm/yr to give a conservative estimation for plate displacement. There are few real constraints

on Archean plate velocities. A few paleomagnetic studies suggest they are similar to those of today (e.g., Brenner et al., 2020), and other models vary from slower (a few), to faster (more models). So, we are being conservative. To address this concern in the text, we add the following reference, and one sentence on lines 524-527.

“We note that paleomagnetic data from the Pilbara craton suggests that Archean plate velocities were likely similar to those of the modern Earth¹⁵, so our conservative estimates are reasonable.”

Brenner, A. R., Fu, R. R., Evans, D. A. D., Smirnov, A. V., Trubko, R., Rose, I. R., 2020. Paleomagnetic evidence for modern-like plate motion velocities at 3.2 Ga. Science Advances 6, eaaz8670. <https://doi.org/10.1126/sciadv.aaz8670>.